# Asking Specifically Instead of Ambiguously to Your GPT Improves Image Caption

## Abstract

The advances in large vision-language models (VLMs) have sparked a growing interest in generating accurate, complete, and user-friendly image captions to enhance downstream multi-modality tasks such as text-to-image generation, text-driven object detection, and grounding. However, current VLM-based image captioning methods often miss important details, recognize incorrect objects or relationships, and deliver suboptimal captions for downstream applications. One primary reason for this issue is the ambiguous prompts typically used, such as "describe this image in detail," which fail to guide the VLM's focus on specific elements within the image. To address this, we extensively explore the difference between using ambiguous prompts and decomposing them into a series of specific questions. We find that asking a series of targeted element-specific questions significantly enhances the attention of VLMs to important objects, the consistency of the answers under repeated questions, and the alignment with their training data distribution. Building on this insight, we introduce ASSIST, a method that systematically decomposes image caption prompts into a sequence of focused questions corresponding to distinct image elements. We annotated 100k images using GPT-4V with this approach and fine-tuned a LLAVA model, resulting in a captioner that greatly improves caption accuracy and quality. Our fine-tuned model recognizes $\times 1.5$ more correct objects and achieves $\times 1.5$ higher precision in describing them on the COCO benchmark compared to vague prompting methods. Additionally, our method produces element-specific answers that can be efficiently organized into graph structures, benefiting tasks like open-vocabulary object detection and image generation. This leads to significant improvements in the accuracy, precision, and mIoU of state-of-the-art detection models, with recall scores increasing by $\times 1.7$ over previous methods. Experiments across diverse scenarios and benchmarks validate the effectiveness of ASSIST. All code, datasets, and models will be made publicly available.

## 1 Introduction

Image captioning is a pivotal task in multi-modal understanding, tasked with generating precise and comprehensive descriptions of images. These descriptions are critical not only for enhancing human-computer interaction but also for facilitating deeper integration between visual data and machine learning models, particularly in applications like visual generation and object detection (Liu et al., 2023; Bai et al., 2023; Chen et al., 2023b; OpenAI, 2023; Podell et al., 2023; Betker et al., 2023). An effective image caption should detail key objects, describe their attributes accurately, and be structured in a user-friendly manner, ensuring seamless utility for downstream models, even those lacking robust text encoders.

Despite their capabilities, even state-of-the-art Vision-Language Models (VLMs) such as GPT-4V often fail to capture all essential elements in their captions, resulting in outputs that lack both accuracy and completeness (OpenAI, 2023). This shortfall is partially due to the inherently ambiguous prompts used in image captioning tasks, which do not specify the desired level of detail or focus, leading to generic and often unhelpful descriptions.

Addressing this challenge, we propose a novel approach that shifts from using ambiguous prompts to posing specific, targeted questions. By decomposing the broad task of captioning into a series of

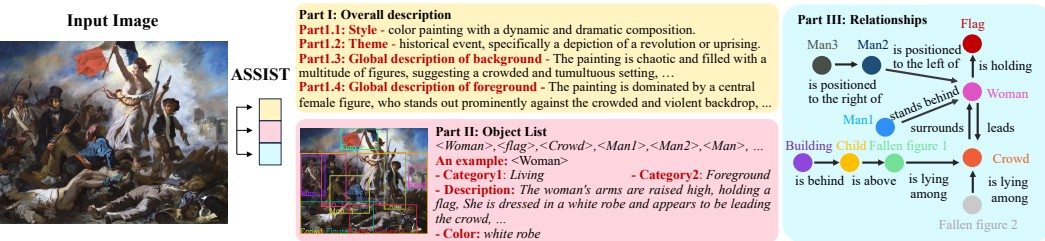

Figure 1: **The ASSIST-style captions** consist of three components: an overall description, an object list, and relationships. Each object in the object list is accompanied by its category information, detailed description, and color information.

specific queries about distinct image elements, we can generate more detailed and accurate descriptions. This method not only improves the granularity of the information captured but also enhances the usability of the captions for both humans and downstream models.

To implement this strategy, we developed a methodological framework called ASSIST (**As**k **S**pecifically **In**stead of Ambiguously **t**o You GPT), which reformulates image captioning into a structured question-answering dialogue. This approach not only identifies more objects by asking specifically about their presence but also elicits more precise attributes by querying details individually. Furthermore, by consolidating responses into a single-turn dialogue using in-context learning techniques, ASSIST reduces the complexity and computational overhead typically associated with multi-turn interactions in VLMs.

Building on this framework, we introduce the ECO (Enumerate Common Objects in Context) dataset, specially curated to train our fine-tuned LLaVA-13B model, referred to as LLaVA(ASSIST)-Captioner. Our evaluation across several multi-modal benchmarks demonstrates significant improvements over baseline models. Additionally, we explore the efficacy of ASSIST-style captions in various downstream applications, including zero-shot visual question answering (VQA), object detection, image generation, and video dense captioning tasks, underscoring the versatility and robustness of our approach.

## 2 BACKGROUNDS

**Image captioning.** Image captioning is an important research topic in the field of artificial intelligence, playing a crucial role in multimodal understanding and image generation. Traditional image captioning methods (Anderson et al., 2018; Mao et al., 2016; Kazemzadeh et al., 2014; Sharma et al., 2018; Vinyals et al., 2015) typically rely on manually annotated datasets, such as MS COCO and Flickr30k, using deep learning techniques to fit the caption datasets. These methods often evaluate performance based on similarity to the dataset. However, limited by the quality of manual annotations, traditional image captioning techniques are gradually being replaced by vision-language models (VLMs) with the rapid advancements in this area.

**Vision language models.** With the rapid development of large language models, vision language models OpenAI (2023) have also advanced quickly. These models typically build upon large language models by introducing a vision encoder based on Vision Transformers (ViTs) (Dosovitskiy, 2020). They often train a simple adapter to align the two modalities (Liu et al., 2023; Bai et al., 2023; Chen et al., 2023b). In addition to this approach, there are models (Team et al., 2023) that do not use adapters and instead directly concatenate visual and textual features, relying on massive datasets and parameter counts to align multiple modalities. The swift progress of VLMs has brought significant innovations to the field of image captioning. Currently, models like GPT-4V excel in image description tasks, significantly outperforming traditional methods based on manually annotated datasets.

**Prompt engineering.** Although VLMs are already quite powerful, they still underperform on certain tasks. In addition to relying on more training to enhance performance, one remarkable aspect of VLMs is their ability to significantly improve results through proper prompt engineering. For instance, Chain of Thought (COT) (Wei et al., 2022) prompts augment the model's reasoning ca-

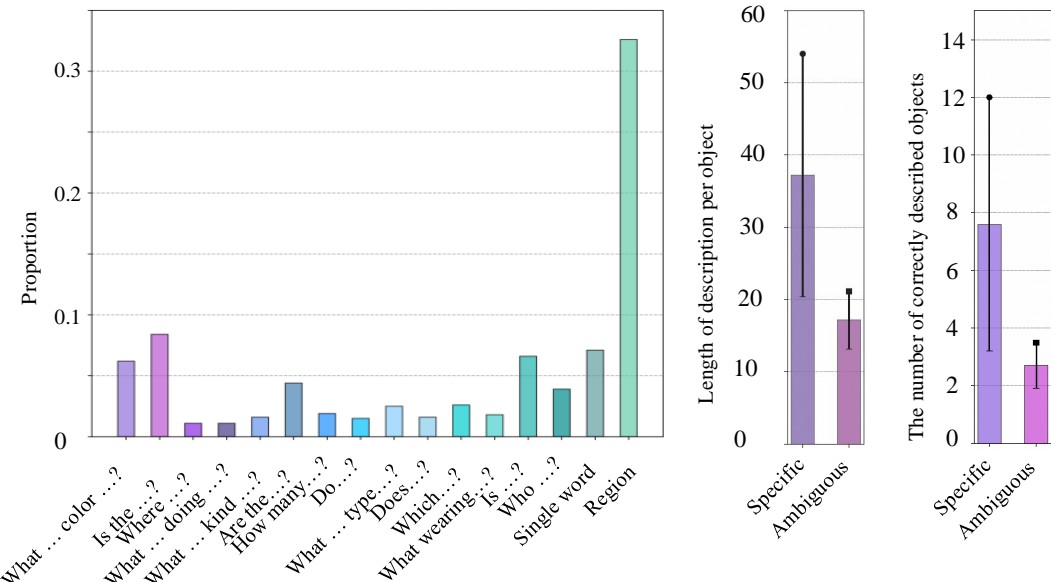

Figure 2: (a) The proportion of specific requests within the training data of LLaVA. (b) Specific requests are more likely to identify correct objects and generate more detailed descriptions.

pabilities by adding intermediate steps in the thought process, which has led to a plethora of related works building on COT (Hu et al., 2023; Yao et al., 2024; 2023; Yu et al., 2023). Moreover, techniques such as Retrieval-Augmented Generation (RAG) (Lewis et al., 2020) and In-Context Learning (ICL) (Brown et al., 2020) are also well-known and highly effective prompt engineering strategies. Similarly, this paper breaks down the abstract image description task into a series of specific requests, thereby greatly enhancing the accuracy, completeness, and user-friendliness of the captions for downstream tasks. This process can also be viewed as a form of prompt engineering.

## 3 THE ASSIST PRINCIPLE: ASK SPECIFICALLY INSTEAD OF AMBIGUOUSLY

Image captioning aims to gather as much correct information as possible from an image and write it down to optimize downstream tasks' performance, like object detection or image and video generation. Yet the current prompting ways seem to have some gap from this target. While the previous methods try to add phases like 'describe in detail' or 'describing the content one can determine confidently' (Chen et al., 2023a) to improve the correctness and completeness of captions, they still struggle to correctly recognize as many as possible existing objects, relations, or other important information. In this paper, we argue that simply adding more pressure to your VLMs, like asking them to be 'super talent image captioners' or 'do not lose anything', may not actually have enough effects in improving results. Rather, what the VLMs still need is specific questions without any ambiguity that can be clearly understood and answered.

So instead of generally asking 'Create detailed captions describing the contents of the given image', we propose to decompose it into a series of specific ones such as 'List all objects in the image', 'Describe the color/attribute/category/location of the first object', 'Describe the first object in detail', or 'Describe the relation between the first object and the second object'.

### 3.1 WHAT MAKES SPECIFIC QUESTIONS BETTER FOR VLMS

In this section, we try to find out *shall VLMs prefer specific questions rather than ambiguous ones when annotating an image?* Yet to answer the above question we have to first identify what is specific questions to a vision language model like ChatGPT or LLaVA. For people, we can say that a specific question should be clearly understood and answered. Generalize this principle to VLMs we can then get what follows: *We say a question is specific for VLMs if*

1. *it can be clearly understood by the VLMs, meaning hidden neurons of VLMs clearly know what you are talking about and are concentrated in the region of interest;*
2. *it can be clearly answered by the VLMs, meaning hidden neurons of VLMs clearly know what the answer is and will give you the same answer whenever you ask.*

With this concept in mind, below we can check whether those specific questions for people raised before are also specific for VLMs and thus are much preferred by the models.

**Clear Understanding** Most VLMs are transformer backbones consisting of several attention blocks. The neurons in those attention blocks tell the focus of the VLMs at the current token. So investigating the attention map between the output tokens and the image token can vividly tell how certainly the VLMs understand the question. As shown in Figure 3, we can observe an outstanding phenomenon that when posed with specific questions, such as enumerating all objects in the image or describing a particular object, the output tokens typically exhibit a stronger correlation with the target region, as evidenced by an enhanced attention map, compared to ambiguous questions like `"Please describe this image."`. It suggests that VLMs understand the meaning of specific questions more firmly than ambiguous ones.

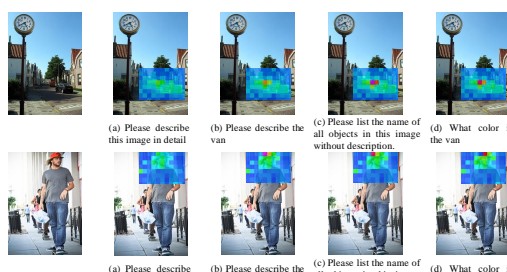

Figure 3: Specific questions result in more pronounced attention (especially crimson points) maps on the target region.

**Clear Answering** We find the specific questions also lead to far more consistent answers under repeated questioning, while ambiguous questions like `'Please describe this image in detail'` can produce very different results even when asking about the same image. We compare this ambiguous question with a decomposed specific question series, consisting of `'Please list the names of the objects in this image.'` followed by `'Please describe {obj} in detail'`. We randomly select 1,000 images from the MSCOCO dataset (Lin et al., 2014) and ask both types of questions to the LLaVA model. For each image, we ask each question 10 times using different random seeds. To calculate answer semantic consistency, we compute the similarity of answer sub-sentences among $n$ different independent repeats of question-answering. Specifically, the Semantic Consistency is computed as

$$
\begin{aligned}
&\text{Semantic Consistency}(a_{i\,i=1}^{n}) \\
&= \sum_{1 \le i \ne j \le n} \frac{1}{2} \left( \text{FitsRatio}(a_i|a_j) + \text{FitsRatio}(a_j|a_i) \right), \\
&\text{FitsRatio}(a_i|a_j) \\
&= \frac{|\{a_i^m \text{ is a sub-sentence of } a_i : \frac{\langle \text{T5}(a_i^m), \text{T5}(a_j^n) \rangle}{\|\text{T5}(a_i^m)\|_2 \|\text{T5}(a_j^n)\|_2} \ge \rho\}|}{|\{a_i^m \text{ is a sub-sentence of } a_i\}|},
\end{aligned}
\tag{1}
$$

where $|\cdot|$ is the counting measure of finite sets, and sub-sentences are sentences split by punctuation. As illustrated in Figure 4, the two specific questions exhibit much higher semantic consistency scores than the ambiguous question, meaning that the VLMs are very confident and clear in what the answers should be.

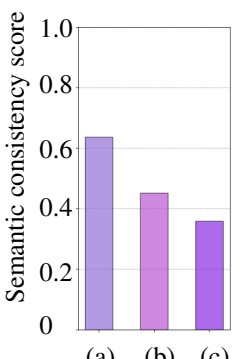

Figure 4: **Semantic consistency scores** of answers to three questions: (a) list all objects; (b) describe an object; (c) describe the image.

**Further Bias from the Train Distribution** The behavior of VLMs largely depends on their training data. Taking LLaVA as an example, we analyze its training dataset and calculate the proportions of specific and ambiguous questions. Our criteria for identifying the data containing specific questions include two main points: 1) Templates that correlate with specific question templates such as `"How many ...?"`, `"What color is ...?"`, `"What time is ...?"` and `"Is there ...?"` (the complete list of used templates can be found in Figure A1); and 2) Responses that consist of only a single word or questions that request a single-word answer. By applying these criteria, we can effectively identify the most specific questions. However, some

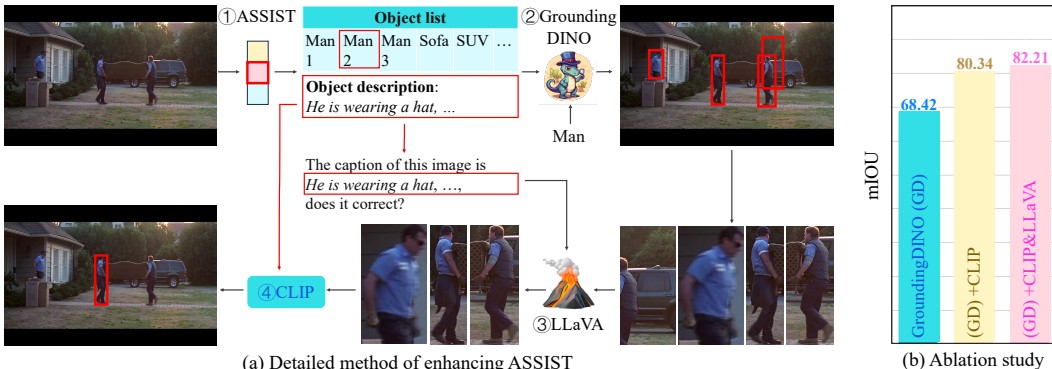

(a) Detailed method of enhancing ASSIST          (b) Ablation study

Figure 5: (a) **Detailed method** for graph grounding. The method contains four steps: 1) Extracting ASSIST from images using GPT-4V or ASSIST-Captioner; 2) Getting candidate regions using Grounding DINO given the name of the object; 3) Using LLaVA to discard blatant incorrect regions; 4) Select the region whose image feature matches the text feature of object description the most by CLIP. (b) **Ablation study** of method in (a), exploring the improvement of introducing CLIP and LLaVA, where the experiment is conducted on ASSIST benchmark.

specific questions may not match the templates and thus could be overlooked. Our analysis reveal that **88.17% of LLaVA's training data could be classified as specific questions**; considering that our matching rules may miss some specific questions, the actual proportion is likely even higher. Given the existence of such data bias, LLaVA is more likely to excel at answering specific questions as it is trained to do so. For other VLMs like GPT-4V or Qwen-VL-Max, although we do not have access to their training data, many of them are fine-tuned using conversational datasets, which suggests that similar bias may also be observed.

**Significant Advantages in Recognizing More Objects** Following the previous settings, we decompose the ambiguous question, "Please describe this image in detail," into several specific questions. We then compare these two questioning methods using GPT-4V and LLaVA as VLMs. For this comparison, we annotate 100 samples using both approaches and manually counted the number of correctly identified objects produced by each method. Additionally, we measure the average length of the descriptions for different objects. The results indicate that in both VLMs, the combination of multiple specific questions leads to image descriptions that significantly identify more objects accurately and provide more detailed information.

## 3.2 Image Caption Using ASSIST Principle

**Design the Specific Question List** The key insight of ASSIST involves decomposing the ambiguous task of image description into a series of concrete sub-tasks. Besides, we need a complete question list to cover all possible information in an image. Borrowing the idea from scene graph, where a graph structural comprising element-wise objects and their relationships (Krishna et al., 2017; Lu et al., 2016; Xu et al., 2017; Johnson et al., 2015; 2018) are commonly used to represent all knowledge in a real-world scenario, we design the sub-tasks following the structure of a scene graph. This includes 1) enumerating all objects within the image and separately describing them, 2) identifying the relationships among these objects, and 3) characterizing the style and themes conveyed in the image.

**Unify Question List into One Prompt** In practice, querying VLMs with specific questions one by one can significantly increase dialogue rounds and thus is deadly expensive in both time and funds. To address this, we propose a method that enables VLMs to answer all questions sequentially within a single dialogue round. Our approach comprises two key steps. First, we design a specific output format that combines answers, using special symbols to separate them for easy parsing. second, we introduce in-context learning (ICL)(Brown et al., 2020) by including hand-crafted examples in the prompt that demonstrate the desired response order and the use of delimiters. In practice, a few simplified examples are sufficient and can be integrated into the prompt, allowing the ICL process

Table 1: **Results of LLaVA(ASSIST)-CAPTIONER on CQA benchmarks.** The QA model is a fixed LLaVA-13B.

| Captioner | NLVR2 | OK-VQA | VQAv1 | VQAv2 |
|---|---|---|---|---|
| ShareGPT-4V-13B | 57.5 | 55.4 | 50.7 | 65.4 |
| Qwen-VL-max | 56.8 | 52.1 | 46.0 | 65.6 |
| LLaVA-13B | 56.3 | 54.8 | 50.0 | 64.1 |
| ASSIST-Captioner | **59.1** | **56.8** | **52.6** | **66.4** |

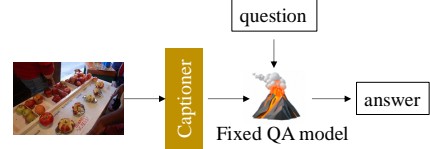

Figure 6: **CQA** for evaluating caption quality, where a fixed QA model answers image-related questions based on the caption of the image instead of the image itself.

Table 2: **Precision & recall scores calculated by manual annotation** between LLaVA(ASSIST)-CAPTIONER and other VLM-based captioners.

| Method | Precision | Recall |
|---|---|---|
| LLaVA | 36.4±1.5% | 59.2±4.7% |
| ShareGPT-4V | 23.2±3.8% | 55.3±2.1% |
| Qwen-VL-max | 35.2±5.9% | 57.5±2.0% |
| GTP4v | 21.5±0.7% | 70.6± 13.4% |
| ASSIST | $\mathbf{56.2 \pm 4.2}$% | $\mathbf{82.8 \pm 8.3}$% |

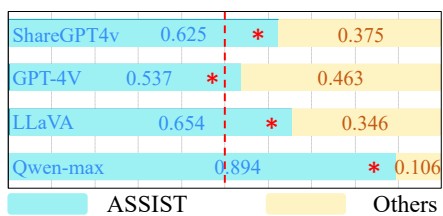

Figure 7: **Win rate of pairwise comparisons** between LLaVA(ASSIST)-CAPTIONER and other VLM-based captioners.

to be completed in just one dialogue round. We use GPT-4V for implementation, with the final instructions detailed in Appendix A.1.3.

**Adding Grounding Capability** ASSIST's structure provides a list of objects required by grounding models, enabling the combination of advanced VLMs for detailing and top-tier grounding models for precise localization within ASSIST. Although Grounding DINO provides accurate object positions, names alone fall short of distinguishing objects within the same category. Here, ASSIST's detailed node descriptions come into play, allowing for precise region identification when used in conjunction with CLIP. Moreover, we enhance grounding accuracy by first applying LLaVA to filter out incorrect bounding boxes before proceeding with the CLIP step. We conducted an ablation study on the ASSIST benchmark (with details in Section 3.2), and the findings, presented in Figure 5 (b), confirm the benefits of incorporating CLIP and LLaVA into our approach. See Figure 5 (a) for an illustration of this process. Based on this approach, we develop the ASSIST dataset, detailed in Section 3.2.

**Collecting the Dataset and Fine-tuning LLaVA** Using the above method, we collected a dataset called **Enumerate Common Objects in Context (ECO)** consisting of 103k image-ASSIST caption pairs. The 100k train split is first annotated using GPT-4V and ASSIST prompt, and then comprehensively re-annotated by human labors to eliminate ambiguities and inaccuracies. The 3k test split, with 27k objects, 148k relationships, is completely annotated manually without preprocessing of VLMs to ensure utmost accuracy and include as many as objects as possible. We then fine-tune a 13B LLaVA model on the train set. The fine-tuned LLaVA model shows similar performance with GPT-4V, having similar distribution of the recognized categories, nouns, and verbs, as is shown in Figure A2. Furthermore, the precision and recall score calculated by manual annotation (the metrics are detailed as Section 4.1.3) show LLaVA(ASSIST)-CAPTIONER achieve 91% of precision score and 90% of recall score of that of GPT-4V. Consequently, LLaVA(ASSIST)-CAPTIONER is a viable alternative to GPT-4V for producing ASSIST and helps us extend ASSIST dataset. The trained LLaVA(ASSIST)-CAPTIONER is also adept at performing additional useful tasks without fine-tuning, such as interactively modifying the items of ASSIST, transforming prompts into ASSIST format, and envisioning scenarios in the style of ASSIST. See details of the dataset and fine-tuned LLaVA in Appendix A.2.

## 4 EXPERIMENTS

In this section, we comprehensively evaluate the image caption produced by applying ASSIST. The evaluation is divided into two parts: directly measuring the caption quality and evaluating the

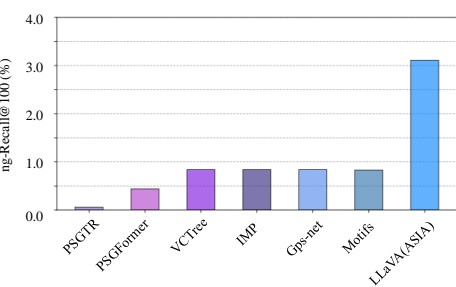
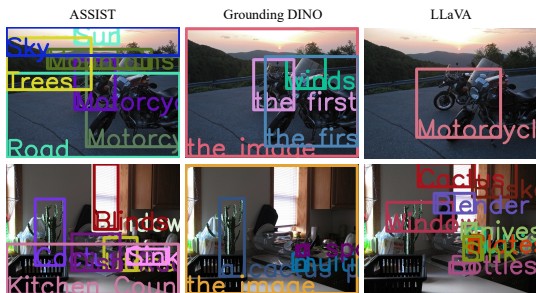

Figure 8: Results of open-vocabulary scene graph generation on six different benchmarks.

Figure 9: Results of zero-shot open vocabulary object detection. ASSIST can correctly recognize significantly more objects.

improvements to downstream zero-shot tasks when applying it. Specifically, we propose a new benchmark adapted from Vision Question Answering (VQA) to directly evaluate the performance of image caption leverage the advances of Large Language Models, which we introduce in detail in 4.1. Considering the limiting and space, implementation details, and introduction to some experiment settings are **placed in Appendix A.5.**

## 4.1 EVALUATING IMAGE CAPTION QUALITY

In this section, we employ three distinct evaluation methods to assess the quality of captions produced by ASSIST. We use LLaVA-13B as our default captioner and denote LLaVA(ASSIST)-CAPTIONER as it fine-tuned version in ECO. These evaluation tasks include our newly proposed evaluation approach, Caption Question Answering (CQA) in Section 4.1.1, and open-vocabulary scene graph generation framework in Section 4.1.2, along with comprehensive user studies in Section 4.1.3.

### 4.1.1 ANALYZING OVERALL QUALITY USING LLMS AND CAPTION QUESTION ANSWERING

**Caption Question Answering (CQA)** The VQA benchmark is broadly used in evaluating the outputs of VLMs. Yet they are not suitable for evaluating captioner models, as 1) the query in VQA benchmarks are very different from image caption, thus they may hardly indicate the ability of image caption of the captioners; 2) captioner models may sacrifice their general VLM capability to enhance image caption ability. So instead, we remove the image in VQA, and replace it with its caption produced by different caption methods. The caption and original query are then sent to an LLaVA-13B model to answer the query purely based on text captions. We call this task the **Caption Question Answering (CQA)**. Considering the tolerable accuracy of LLaVA-13B model, the right or wrong of CQA is then purely decided by the image caption quality of captioners.

We compare our model against three advanced VLM-based captioners, namely LLaVA-13B (Liu et al., 2023), Qwen-VL-max (Bai et al., 2023), and ShareGPT-4V (Chen et al., 2023a) across four benchmarks: NLVR2 (Suhr et al., 2018), VQAv1 (Antol et al., 2015), VQAv2 (Goyal et al., 2017), and OK-VQA (Marino et al., 2019) within the CQA setting. The results demonstrated in Table 1 indicate that our approach exhibits significant advantages, underscoring that LLaVA(ASSIST)-CAPTIONER generates captions containing much more correct and useful information.

### 4.1.2 ANALYZING OBJECT & RELATION ACCURACY USING OPEN-VOCABULARY SCENE GRAPH GENERATION

We then try to find benchmarks to measure the accuracy of object and relation recognition of the proposed method. Since an image can often be explicitly represented using a scene graph composed of objects and their relationships (Krishna et al., 2017; Lu et al., 2016; Xu et al., 2017; Johnson et al., 2015; 2018), we can utilize the open-vocabulary scene graph generation (OV-SGG) benchmark, which aims to identify (subject-predicate-object) triplets in images, to evaluate the performance of LLaVA(ASSIST)-CAPTIONER. The details of implementing this evaluation are provided in Appendix A.5.1. We compare the performance of LLaVA(ASSIST)-CAPTIONER against several

Table 3: **Comparison of open-vocabulary object detection** among ASSIST, Grounding DINO, open-vocabulary object detection models, and grounding caption models on ASSIST benchmark. We have calculated error bars for models that exhibit randomness.

| Method | AP50($\uparrow$) | Recall($\uparrow$) | mIOU($\uparrow$) |
|---|---|---|---|
| OV-DQUO | 4.7 | 10.7 | 66.5 |
| DE-VIT | 19.3 | 23.8 | 76.8 |
| Grounding DINO | 33.1$\pm$2.5 | 20.2$\pm$0.1 | 75.7$\pm$0.1 |
| Next-Chat | 29.1$\pm$0.1 | 7.7$\pm$0.1 | 67.1$\pm$0.0 |
| Kosmos-2 | 34.2$\pm$4.8 | 13.3$\pm$2.4 | 76.1$\pm$0.4 |
| GLaMM | 34.3 | 19.8 | 79.6 |
| ASSIST | **37.7 $\pm$ 0.9** | **35.9 $\pm$ 0.7** | **79.9 $\pm$ 0.1** |

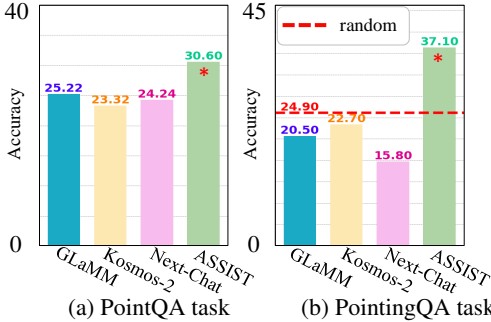

Figure 10: **Quantitative comparison on (a) PointQA and (b) PointingQA** between ASSIST and baselines.

specialized SGG methods, including Motifs (Zellers et al., 2018), GPS-Net (Lin et al., 2020), VC-Tree (Tang et al., 2019), PSGTR, PSGFormer (Yang et al., 2022), and IMP (Xu et al., 2017), using the widely recognized Visual Genome benchmark (Krishna et al., 2017). The results presented in Figure 8 demonstrate that LLAVA(ASSIST)-CAPTIONER significantly outperforms the previous SGG models, thereby validating the high quality of its output captions.

### 4.1.3 ANALYZING PRECISION & RECALL USING USER STUDY

We conduct a user preference study to examine the precision and recall score by manual annotation. We compare LLAVA(ASSIST)-CAPTIONER with LLaVA, Qwen-VL-max, ShareGPT-4V, and GPT-4V, by analyzing captions produced for a randomly sampled set of 200 images from the MSCOCO dataset (Lin et al., 2014). We engage 10 human annotators for manual labeling. For the precision and recall scores, we first extract all important nouns existing in the captions (see details in Appendix A.5.5) and then ask annotators to count the number of objects in the image and the number of correct predictions in the extracted nouns. Then, the precision and recall score can be calculated. In the user preference study, annotators select their preferred annotation in pairwise comparisons, ensuring structural aspects are neutralized to prevent any biases. The outcomes, as shown in Figure 7 and Table 2, indicate ASSIST outperforms all comparisons in general, **notably predicting more correct objects than multiple popular VLMs even containing GPT-4V**.

### 4.2 EVALUATING CAPTION QUALITY USING DOWNSTREAM TASKS IN ZERO-SHOT SETTINGS

For downstream models that lack a large language model (LLM) as a text encoder, comprehending complex image annotations generated by VLMs becomes a notably challenging task. Fortunately, the adaptable structure of ASSIST enhances downstream models' ability to comprehend complex text and empowers them to perform tasks that were previously beyond their capabilities. In this section, we evaluate performance across four downstream tasks. These include improving grounding DINO for open-vocabulary object detection (OVD) as described in Section 4.2.1, enhancing LLaVA for zero-shot point question answering (ZS-PointQA), and zero-shot pointing question answering (ZS-PointingQA) in Section 4.2.2, boosting SDXL for image generation, detailed in Section 4.2.4, and enhancing SAMv2(Ravi et al., 2024) for automated multi-object video tracking while seamlessly extending to the task of dense video captioning, as discussed in Section 4.2.3. **It is important to emphasize that all experiments in this section are conducted within the zero-shot setting.**

### 4.2.1 OPEN-VOCABULARY OBJECT DETECTION

Grounding DINO can process image descriptions to detect mentioned nouns within corresponding images. However, its limited text comprehension often leads to hallucinations, resulting in the identification of irrelevant objects and difficulty distinguishing between instances of the same category (as illustrated in Appendix A.5.1). The ASSIST-style caption mitigates these issues by replacing standard image descriptions with a format that allows Grounding DINO to use item names from the

| Input Prompt | ASSIST + SDXL | DALL-E 3 | SDXL |
|---|---|---|---|
| The full moon hangs high on the left side of the sky, with stars dotted around it. The background on the left side of the picture is a magnificent castle, standing on a distant hill. In the bottom left corner of the photo, two people are standing hand in hand on a rock by the river. On the right side of the picture is a male angel holding a trumpet and spreading his wings. Above the angel, there is a glowing flying saucer suspended in the air. | | | |

Figure 11: **Comparative examples of image generation** reveal that LLAVA(ASSIST)-CAPTIONER enhances advanced generative models like SDXL. SDXL and DALL-E 3 struggle with complex text and fail to produce corresponding images. Remarkably, ASSIST not only elevates SDXL's image quality but also markedly boosts its comprehension of intricate instructions, enabling it to surpass DALL-E 3 in terms of accurately generating images aligning with textual directives.

object list as grounding prompts. It can also utilize object descriptions with CLIP (Radford et al., 2021) to accurately locate target objects.

**Evaluation.** Traditional evaluation on OVD task typically categorizes the dataset into base and novel classes, training on base classes and evaluating on datasets with novel classes. This resembles a zero-shot rather than an open-vocabulary setting, given the finite number of categories (for example, 80 in COCO). To better reflect an open-vocabulary setting, we evaluate the OVD task on the test benchmark of ECO. Instead of using traditional detection metrics (such as AP50, recall, and mIoU) directly, we modified these algorithms to utilize CLIP similarity between predictions and ground truth for label matching. A successful match is established when the similarity exceeds a predefined threshold without requiring complete correspondence between the prediction and ground truth. In addition to OVD methods including OV-DQUO (Wang et al., 2024) and DE-VIT (Zhang et al., 2024), we also compare the performance with grounding caption models, including GLaMM (Rasheed et al., 2024), Kosmos-2 (Peng et al., 2023), Next-Chat (Zhang et al., 2023). **Results.** Results in Table 3 show that ASSIST enhances grounding DINO to outperform all evaluated methods on the test benchmark of ECO.

### 4.2.2 ZERO-SHOT POINTQA AND ZERO-SHOT POINTINGQA

Zero-shot PointQA (Mani et al., 2020) requires VLMs to answer questions about target regions based solely on the provided image caption rather than the image itself. Similarly, zero-shot PointingQA (Zhu et al., 2016) involves VLMs selecting a relevant region from a set of candidates, also relying exclusively on the image caption. These tasks are particularly challenging for VLMs that have not been fine-tuned for such purposes as LLaVA. However, the ASSIST-style captions provide a list of objects along with their corresponding regions and detailed descriptions. Leveraging this information allows us to extract region-relevant descriptions that can effectively address both the zero-shot PointQA and PointingQA tasks. We outline the specific methodology for utilizing AS-SIST-style captions to assist in these tasks in Appendices A.5.2 and A.5.3. **Evaluation.** We compare our model with grounding caption models, including GLaMM, Kosmos-2, and Next-Chat, that can simultaneously obtain corresponding object bounding boxes from captions on both tasks. Specifically, we evaluate the zero-shot PointQA task on LookTwice-QA dataset (Mani et al., 2020) (as shown in Figure 10 (a)) and the zero-shot PointingQA task on Visual-7W dataset (Zhu et al., 2016) (as illustrated inFigure 10 (b)). The results demonstrate that our LLAVA(ASSIST)-CAPTIONER outperforms advanced grounding caption models in both tasks.

### 4.2.3 MULTI-OBJECT VIDEO TRACKING AND DENSE VIDEO CAPTIONING

The advanced segmentation model SAM-2 (Ravi et al., 2024) supports stable video tracking and holds significant potential for dense video captioning. However, SAM-2 requires text prompts or indicator points to accurately locate target objects, complicating its direct application in video captioning. Fortunately, the ASSIST-style caption addresses this gap. Specifically, the ASSIST-style caption includes an object list that specifies the targets for tracking. Additionally, since each object is associated with a mask, the centroids of these masks can serve as effective indicator points. Finally, the detailed annotations within the ASSIST-style caption provide descriptions that extend beyond the capabilities of SAM-2. An example illustrated in Figure A14 demonstrates how this ap-

proach efficiently captures the continuity and evolution of video content, resulting in coherent and descriptive narration. Additional examples can be found in Appendix A.5.6.

### 4.2.4 IMAGE GENERATION

Advanced text-to-image generative models like SDXL (Podell et al., 2023) struggle to follow complex text prompts and accurately generate images. Fortunately, ASSIST-style caption allows generative models to split the challenge into three easy parts: **1) step1.** Given a natural prompt for generation, the trained LLAVA(ASSIST)-CAPTIONER can transform it into the ASSIST-style caption along with a plan of the positions for all objects (discussed in Appendix A.4). **2) step2.** Utilizing the background description part and the detailed description of each object in the object list part, SDXL can generate the background and the important objects separately. **3) step3.** Those generated parts are merged according to the planned positions in step1, and then the

Table 4: **Accuracy in depicting objects ($A_o$) and relationships ($A_r$) in images generated from text prompts**, as evaluated by human. We compare SDXL enhanced by LLAVA(ASSIST)-CAPTIONER with SDXL and DALL-E 3.

| Method | $R_o(\uparrow)$ | $R_r(\uparrow)$ |
|---|---|---|
| SDXL | 59.2±4.0% | 41.5±3.5% |
| DALL-E 3 | 90.1±4.2% | 71.6±3.4% |
| ASSIST + SDXL | $\mathbf{95.2 \pm 1.1\%}$ | $\mathbf{76.7 \pm 0.9\%}$ |

final image can be refined by common refine methods such as inpainting (Rombach et al., 2022) and SDEdit (Meng et al., 2021). **Evaluation.** To quantitatively assess the correlation between the text prompts and the generated images, we conduct a user study involving 10 human annotators and 100 samples. They are required to first annotate the significant objects and relationships mentioned in the text prompts and then count the number of correctly generated ones in the images. Thus we can compute the recall metrics for objects ($R_o$) and relationships ($R_r$). As detailed in Table 4, the results demonstrate that ASSIST-style caption significantly enhances SDXL's ability to understand and follow complex prompts. Remarkably, it enables SDXL to surpass DALL-E 3 in faithfully reproducing the details specified in the text descriptions. This conclusion can be further supported by the quantitative examples shown in Figure 11 (more instances available in Appendix A.5.4).

## 5 CONCLUSION

In this work, we addressed the limitations of current VLM-based image captioning methods, which often fail to capture critical details and relationships, resulting in suboptimal performance for downstream tasks. Through extensive exploration, we demonstrated that ambiguous prompts like "describe this image in detail" do not provide sufficient guidance for VLMs to focus on important elements in images. To overcome this, we proposed ASSIST, a method that decomposes image caption prompts into a sequence of specific, element-focused questions, significantly enhancing the model's ability to recognize and describe objects accurately. By annotating 100k images and fine-tuning a LLAVA model, our approach resulted in substantial improvements in both caption quality and precision. Our method consistently outperforms vague prompting techniques, achieving a ×1.5 improvement in object recognition and precision on the COCO benchmark. Additionally, the structured, element-specific answers generated by ASSIST benefit other tasks, such as open-vocabulary object detection and image generation, leading to a ×1.7 increase in precision and significant boosts in mIoU for detection models. These findings validate the effectiveness of ASSIST in enhancing VLM-based captioning and its applicability to various multimodal tasks.

## 6 LIMITATIONS

This paper introduces a method designed to assist smaller models in comprehending complex texts and to facilitate their integration with VLMs, achieving remarkable performances across multiple benchmarks. However, despite these achievements, our approach still faces certain limitations. Firstly, given the absence of a fully automated method that guarantees reliable quality, our data collection process still necessitates human annotation involvement. Secondly, due to cost and resource constraints, the captioner's localization capabilities remain insufficient, necessitating the combination of a grounding model to obtain high-quality positional information.

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

Table A1: **Results of ASSIST-13B on 9 general visual-language benchmarks.** ∗ denotes that the training images of the datasets are observed during training.

| Model | VQA$^{v2}$ | GQA | VizWiz | SQA$^I$ | VQA$^T$ | POPE | MMB | MMB$^{CN}$ | SEED | MM-Vet |
|---|---|---|---|---|---|---|---|---|---|---|
| BLIP | 41.0 | 41.0 | 19.6 | 61.0 | 42.5 | 85.3 | - | - | 46.4 | 22.4 |
| InstructBLIP-13B | - | 49.5 | 33.4 | 63.1 | 50.7 | 78.9 | - | - | - | 25.6 |
| Shikra | 77.4∗ | - | - | - | - | - | 58.8 | - | - | - |
| IDEFICS-80B | 60.0 | 45.2 | 36.0 | - | 30.9 | - | 54.5 | 38.1 | - | - |
| Qwen-VL | 78.8∗ | 59.3∗ | 35.2 | 67.1 | 63.8 | - | 38.2 | 7.4 | 56.3 | - |
| Qwen-VL-Chat | 78.2∗ | 57.5∗ | 38.9 | 68.2 | 61.5 | - | 60.6 | 56.7 | 58.2 | - |
| LLaVA-13B | 80.0 | 63.3 | 53.6 | 71.6 | 61.3 | 85.9 | 67.7 | 63.6 | 61.6 | 35.4 |
| VILA-13B | 80.8 | 63.3 | 60.6 | 73.7 | **66.6** | 84.2 | 70.3 | 64.3 | 62.8 | 38.8 |
| ASMv2-13B | 81.0 | 63.9 | 58.1 | 87.1 | 60.2 | 86.3 | 74.4 | 64.3 | 66.3 | 41.3 |
| ASSIST-13B (ours) | 80.8 | 63.5 | 57.1 | **91.3** | 59.5 | **88.0** | 74.6 | **68.2** | 65.9 | **41.6** |

# A  APPENDIX

The appendix is divided into four sections. It begins with additional methodological details in Appendix A.1. Next, Appendix A.2 covers the specifics of training the ASSIST-VLMs. The following section, Appendix A.3, focuses on human annotation aspects during the dataset collection process for ASSIST. Finally, Appendix A.5 provides a comprehensive overview of the experimental setup, including the metrics used and the methodology for implementing ASSIST in downstream tasks, along with supplementary experimental findings.

## A.1  SUPPLEMENTARY METHODOLOGICAL DETAILS FOR ASSIST

In this section, we provide four key details about the ASSIST method. First, we present several complete examples of ASSIST-style captions, as illustrated in Figures A7 and A8. Second, in Appendix A.1.1, we display the complete templates for identifying specific questions mentioned in Section 3.1. Third, in Appendix A.1.2, we provide the complete list of specific questions metioned in Section 3.2, along with the corresponding answer templates. Finally, we showcase and analyze the complete prompts used to generate ASSIST-style captions from VLMs in Appendix A.1.3.

### A.1.1  COMPLETE TEMPLATES FOR IDENTIFYING SPECIFIC QUESTIONS

In Section Section 3.1, we analyze the different performances of LLaVA in answering specific questions versus abstract questions by examining its training data. A key point of our analysis involves the automated identification and quantification of specific questions from over 3 million training conversations. We apply two main principles for this process.

First, we classify the questions of those conversations whose answers are overly brief consisting of a single word or whose questions require a single-word answer as specific questions. This is because abstract questions usually elicit more varied responses that cannot be easily summarized in one word; a single word response typically corresponds to yes/no, numeric, or simple noun answers, which are characteristic of specific questions.

Second, we designed a series of templates for specific questions to facilitate string matching. The templates include three main matching patterns:

- The questions starting with special words such as `"Who"`, `"where"` and `"how many"`. These questions typically lead to specific answers. For example, `"Who"` generally refers to a person, `"Where"` to a location, and `"How many"` to a numeric response.

- `"What"` combined with specific terms like `"what ... color ..."` which specifically asks about color, `"what ... time ..."` focus on time inquiries, and `"what ... type ..."` which targets categories. We chose this approach because while `"what"` can lead to abstract questions, it can also direct queries toward specific details. Various phrasings can ask about color, such as `"what color is ..."` or `"what is the color of ..."`, allowing us to check for keywords like "color" when the question begins with `"what"`.

| | | | |
|---|---|---|---|
| 1. What … doing ... | 18. What … position ... | 35. Where... | 49. What is the |
| 2. What … holding … | 19. What … setting … | 36. Which … | occupation … |
| 3. What … appearance … | 20. What … condition … | 37. Who is … | 50. What is the main |
| 4. What … hanging … | 21. What … placed … | 38. Who … | feature … |
| 5. What … color … | 22. What … size … | 39. Are … | 51. Does … |
| 6. What … wearing … | 23. What … gender … | 40. Is … | 52. Do … |
| 7. What … expression … | 24. What … material … | 41. What object is … | 53. In what type … |
| 8. What … wearing … | 25. What … action … | 42. What furniture… | 54. Has … |
| 9. What … type … | 26. What … made of … | 43. What animal … | 55. Have… |
| 10. What … kind … | 27. How many … | 44. What activity … | |
| 11. What … color … | 28. How much … | 45. What is the main | |
| 12. What …  whether … | 29. How large … | object … | |
| 13. What …  time … | 30. How full is … | 46. What is the primary | |
| 14. What … currency … | 31. Are the … | object … | |
| 15. What … brand … | 32. Is the … | 47. What is next … | |
| 16. What … theme … | 33. Is there … | 48. What accessories … | |
| 17. What … locate … | 34. Are there … | | |

Figure A1: **All the requests template used to identify the specific questions**.

- Location-related queries: If a question includes terms like `"region"` or bounding box, it indicates that the query targets a specific area in the image or requests a bounding box prediction. Such characteristics clearly identify specific questions.

We provide a detailed list of all templates in Figure A1. Our experiments reveal that, based solely on single-word answer or single-word requirement of answer, approximately 48.80% of the questions are identified as specific questions. Relying solely on template matching, this percentage is 76.73%. When both methods are combined, as mentioned in the main paper, 88.16% of the questions are recognized as specific questions. It's important to note that while these matching strategies ensure precision, they do not guarantee recall; due to the diversity in responses and questions, it is impossible to find all templates and we are likely to miss many specific questions. Hence, the actual proportion of specific questions in LLaVA's dataset may well exceed 88.16%.

### A.1.2    THE FORMAT OF ASSIST-STYLE CAPTION IN STRING FORMAT

As discussed in Section 3.2, we design a specific output format for VLMs that sequentially combines the answers to different questions. In this section, we introduce the string format, where an example is shown in Figure A9. Specifically, we label main titles with %% and subtitles with &&. When listing objects, we enclose extra details like category, description, and color in brackets (). Each detail is separated by a semicolon ";". We mark the name of an object with <>. During the description of relationships, we use <> for showing objects and [] for the predicate. Additionally, we use <> to highlight important objects within the object, serving multiple purposes. One such function is to post-process the GPT-4V output results. This involves removing foreground information from the background description by deleting sentences where the foreground objects appear, or similarly, eliminating background information from the foreground description. By using these special symbols to separate different sections, we can effortlessly organize the string format output of VLMs into a ASSIST-style caption using regular expressions. This makes it easy for downstream tasks to extract various pieces of information without any hassle.

### A.1.3    INSTRUCTION FOR GPT-4V TO OBTAIN ASSIST-STYLE CAPTIONS

An example of the final question prompt can be seen in Figure A10. As discussed in Section 3.2, we implement the ICL technique to force VLMs to respond in the desired order and use the special delimiter symbols. In practice, we discover that GPT-4V does not require exhaustive examples to master the desired format. We simply need to insert a few important examples in the right spots within the instruction, which then play a key role. You can see the final instruction in Figure A10, where we've highlighted the critical examples in orange. Among the examples used, some are specific and others are more general. We've observed that for straightforward structural elements,

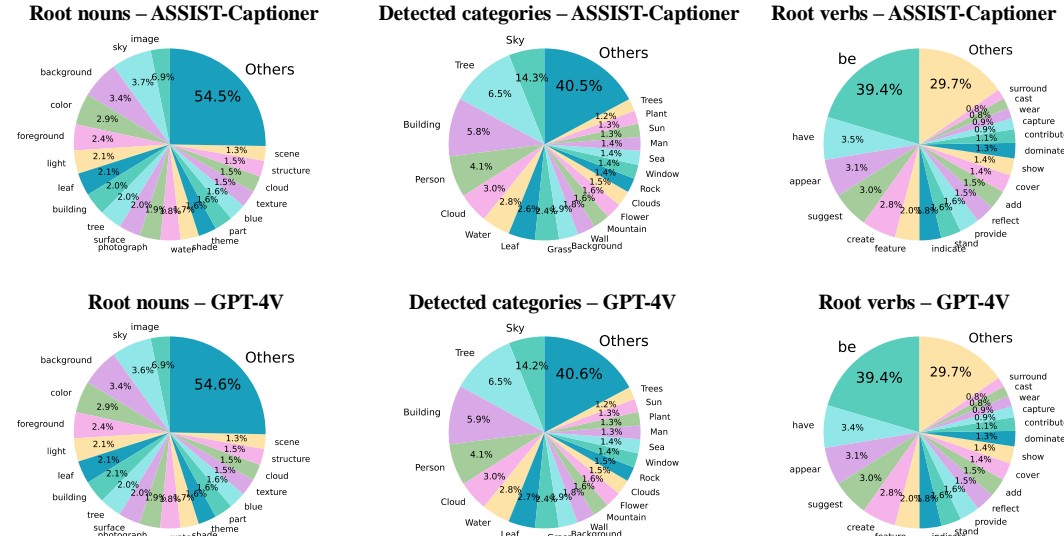

Figure A2: **Analyzing the root words and detected categories in ASSIST's output on testset**: We compare the root words and detected categories generated by LLaVA(ASSIST)-Captioner and GPT-4V, with certain sections magnified for clearer visualization. The results reveal that the output distribution of ASSIST closely resembles that of GPT-4V.

general examples are quite effective. For instance, just a few lines, like 'lines 3-4' or 'lines 8-9', can adequately indicate the use of special symbols in a section, eliminating the need for a full-fledged example. In lines 21-22, we present a general example that clearly delineates the structure of each object, which significantly minimizes GPT-4V's errors. To keep object details easy to grasp, we use a general example lines 23-24, which are sufficient for producing simple sentences. Regarding lines 27-28, a general example is enough to instruct GPT-4V on the basic pattern for depicting relationships. Lastly, a general example set out in line 29 aids in preventing GPT-4V from repeatedly generating two-way relationship pairs.

However, our high demands on the content and structure are extremely hard even for GPT-4V. Therefore, GPT-4V sometimes gets details wrong, like missing special symbols, even when we use general examples. That's why we need to use specific examples to make sure GPT-4V really gets the structure. Take numbering items in the same category, for instance, we introduce a specific example in lines 14-15. Without this example, GPT-4V tends to forget to number the items correctly, even though we've already required it in lines 13-14. Also, we noticed GPT-4V does well with the format of the first section but often slips up with the second and third parts, which complicates turning the data into a dictionary. By providing only one clear example for these sections, GPT-4V is much more likely to produce the right structure. The ICL technique has helped ensure that nearly all of the 110k data entries we've gathered are formatted correctly and can be translated into a dictionary format.

## A.2 DETAILS ON LLaVA(ASSIST)-CAPTIONER

**LLaVA(ASSIST)-CAPTIONER as an effective alternative to GPT-4V on captioning task.** We show the analysis of the root words and categories detected in the outputs of LLaVA(ASSIST)-CAPTIONER, which can be seen in Figure A2. The result clearly shows that the output pattern of LLaVA(ASSIST)-CAPTIONER is very close to that of GPT-4V. Notably, there's a 100% overlap in the top 100 frequent nouns, 99% for verbs, and 97% for categories detected by GPT-4V and LLaVA(ASSIST)-CAPTIONER. This similarity confirms that LLaVA(ASSIST)-CAPTIONER can effectively take over from GPT-4V in generating ASSIST from images and extend our ASSIST dataset.

**Training.** LLaVA(ASSIST)-CAPTIONER is fine-tuned on ASSIST training dataset from a pre-trained 13B LLaVA model using Low-Rank Adaptation (LoRA) Hu et al. (2021) technique. The

Table A2: **Complete hyper-parameters** of training ASSIST-13B and LLaVA(ASSIST)-CAPTIONER.

| ASSIST-13B | | | | ASSIST-Captioner | | | |
|---|---|---|---|---|---|---|---|
| Hyper-parameter | Value | Hyper-parameter | Value | Hyper-parameter | Value | Hyper-parameter | Value |
| Lora rank | 128 | Learning rate | $1 \times 10^{-4}$ | Lora rank | 128 | Learning rate | $2 \times 10^{-4}$ |
| Epochs | 1 | Warmup ratio | 0.03 | Epochs | 3 | Warmup ratio | 0.03 |
| Batch size | 128 | Max length | 2048 | Batch size | 16 | Max length | 2048 |

number of LoRA parameters is around 0.5B. The captioner is trained on NVIDIA A100 GPUs, taking around 100 GPU hours.

We provide the hyper-parameters of both ASSIST-13B and LLaVA(ASSIST)-CAPTIONER in Table A2 for better reproduction.

### A.3 COLLECTION OF ASSIST DATASET

As we've mentioned in Section 3.2, creating the ASSIST dataset's training and test sets involves human annotations.

**Collecting training data.** In the process of collecting training data, ASSIST significantly reduces the workload of annotation. It breaks down the complex descriptions into basic elements, for many of which annotators simply need to make a straightforward judgment of right or wrong, a task that is remarkably simple. For large pieces of information such as background or foreground descriptions, annotators are asked to separately determine if each sentence is correct according to the image. Besides, the annotators are asked to add objects missed by GPT-4V. In this process, the structure we designed for objects can help annotators simplify the description process. They only need to fill in the corresponding information according to the structure.

**Collecting test benchmark.** In the method of collecting the test set of ASSIST, annotators are involved in four parts. For the first part, they are expected to correct the result returned by VLMs to recognize the object name given the masked image. In the second and third parts, annotators are asked to separately determine if each sentence is correct. They don't have to add objects as Segment anything (SAM) Kirillov et al. (2023) in this method has ensured that there will be no omissions. At the last stage, they have to determine if a relationship is correct and add an important relationship omitted by VLMs.

### A.4 IMPRESSIVE CAPABILITIES OF LLaVA(ASSIST)-CAPTIONER

In addition to generating ASSIST-style captions from images, the trained LLaVA(ASSIST)-CAPTIONER excels in several additional functions, including interactively editing ASSIST-style captions by requesting desired changes from the LLaVA(ASSIST)-CAPTIONER, transforming ordinary prompts into ASSIST-style captions, and planning the positions of objects within the object list. First, as illustrated in Figure A5, **the LLaVA(ASSIST)-CAPTIONER enables interactive editing of ASSIST-style captions**, thereby influencing the image generation process. Besides and remarkably, without requiring any fine-tuning, **the LLaVA(ASSIST)-CAPTIONER can convert a standard prompt into a ASSIST-style caption**. This capability is particularly important for image generation, given the challenges of manually providing ASSIST-style prompts. We present examples of this functionality in Figure A5. Furthermore, **the LLaVA(ASSIST)-CAPTIONER can effectively arrange the positions of objects within the object list**. Examples of both expanding and organizing prompts can be found in Figures A11 and A12. We quantitatively evaluate the planning capabilities of the LLaVA(ASSIST)-CAPTIONER against LayoutGPT (Feng et al., 2024) on the MSCOCO dataset (Lin et al., 2014) and our ASSIST datasets, employing mIoU, precision, and recall metrics (Feng et al., 2024), as detailed below. The results presented in Table A3 demonstrate that the LLaVA(ASSIST)-CAPTIONER outperforms LayoutGPT across both evaluated datasets.

**Evaluation metrics.** Evaluating the performance of the planning task is a subject that hasn't been widely discussed. As one of the pioneers, LayoutGPT (Feng et al., 2024) collected some images

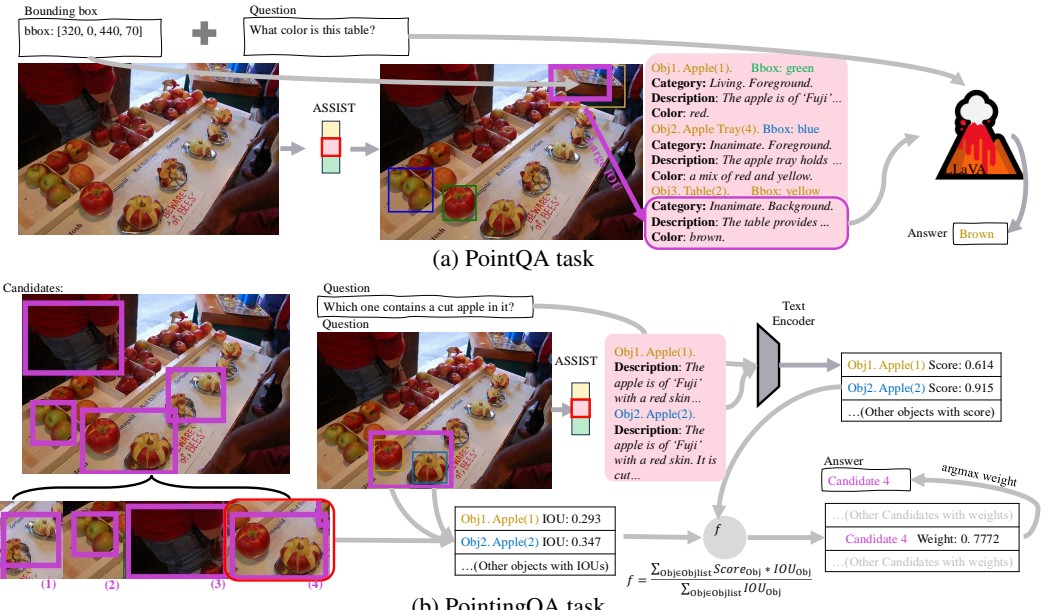

(a) PointQA task

(b) PointingQA task

Figure A3: **An illustrative diagram depicting how ASSIST aids downstream models** in executing PointQA and PointingQA tasks. In (a) the PointQA task, a list of objects and their corresponding descriptions provided by ASSIST are utilized. The description of the object with the large overlap with the target region is used to represent the description of that region; this regional description is then fed into a QA model to answer questions related to the region. In (b) the PointingQA task, object descriptions provided by ASSIST are used to calculate similarity scores with the input question, generating scores for each object. Based on the overlap between object positions and candidate regions, a weighted sum of all object scores is computed to assign scores to candidate regions; the region with the highest score is then selected as the prediction.

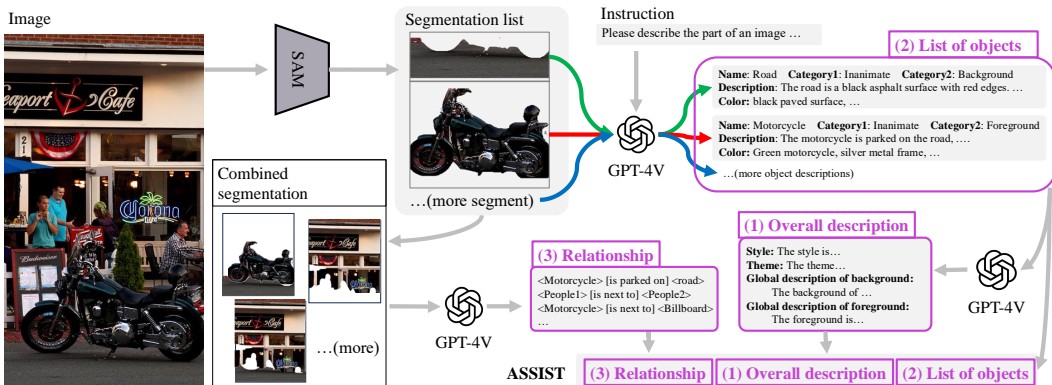

Figure A4: **A detailed overview of the method used to collect the ASSIST benchmark**, segmented into five distinct steps. 1) The SAM model segments all components within the image. 2) VLMs identify the names of objects in the masked image obtained from the first step. 3) Using the names identified in the second step, VLMs annotate each object in detail. 4) VLMs generate an overall description of the image based on the list of objects derived from the above steps. 5) images created by randomly pairing two masked images from the first step are fed to VLMs to identify the relationship between the combined segments. It is important to note that human annotation is required to correct and verify the outputs from steps two through five.

from the COCO dataset (Lin et al., 2014), which have varying numbers of objects of the same category and used precision and recall as evaluation metrics to assess whether the quantity of objects planned is accurate. Inspired by their approach, we have slightly expanded the concepts of precision and recall. We randomly sample 1000 images from COCO and use their official captions as input for

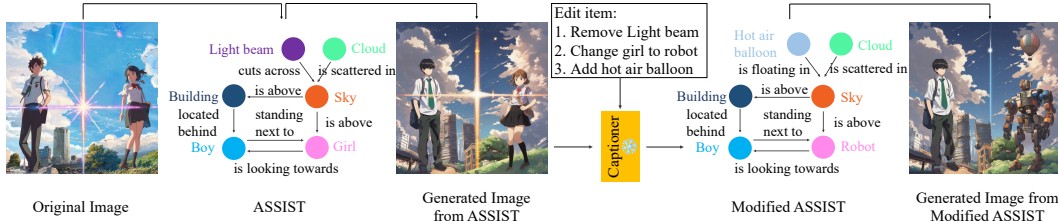

Figure A5: **An example of interactively modifying ASSIST** using LLAVA(ASSIST)-CAPTIONER.

Table A3: **Comparison of plan task** between LLAVA(ASSIST)-CAPTIONER and Layout-GPT (Feng et al., 2024) on both MSCOCO and test benchmark of ECO.

| Dataset | Method | Precision | Recall | mIOU |
|---|---|---|---|---|
| MSCOCO | LayoutGPT | 70.1% | 39.7% | 4.1% |
| | ASSIST | **71.2%** | **41.8%** | **6.8%** |
| Bacon-Dataset | LayoutGPT | 50.8% | 29.2% | 9.1% |
| | ASSIST | **51.7%** | **47.1%** | **18.4%** |

either LayoutGPT or LLAVA(ASSIST)-CAPTIONER. Then, we apply precision and recall metrics to assess how many of the objects predicted by different planning methods actually exist in the images, and how many objects present in the images are predicted.

It's important to note that both the captioner and LayoutGPT operate in an open-vocabulary manner. Hence, we used CLIP to map the open-vocabulary predictions to COCO's fixed set of categories. Specifically, for an open-vocabulary prediction, we compute its similarity to all categories in COCO, treating the similarity as logits, and then use a softmax function to map it to a category in COCO. If the softmax score for the most likely category exceeds a threshold (0.9 here), we consider the prediction to be correct; otherwise, it is deemed incorrect. In ASSIST dataset, the situation is quite similar. A slight difference is that the model's predictions are mapped onto the list of ground truth objects for the current image, rather than a fixed set of categories. Similarly, when the softmax score exceeds a certain threshold, it is considered a correct prediction. Given that ASSIST benchmark is significantly more challenging than COCO, if the threshold is set too high, almost all predictions would be incorrect; hence, we lowered the threshold to 0.5.

Precision and recall do not take into account the positioning of the planning. This is because evaluating whether a position is appropriate is a subjective task, and so long as it is reasonable, it should suffice. Nonetheless, since the positional distribution in the original images is assuredly reasonable, we can also use the positions in the original images as a certain reference. Therefore, we calculated the mean Intersection Over Union (mIOU) of the positions of the objects in the planning compared to those in the original images, and used this as an evaluation metric.

### A.5 SUPPLEMENTARY OF EXPERIMENTS

In this section, we provide supplementary explanations for the experimental details omitted in the main text (Section 4), including the training details of LLAVA(ASSIST)-CAPTIONER, the specific manner in which ASSIST aids downstream tasks, the exact calculation methods for metrics, and any special processing applied to the datasets. We will organize this section following the structure of the main text (Section 4) to facilitate readers in quickly locating the corresponding section for each experiment.

#### A.5.1 OPEN-VOCABULARY OBJECT DETECTION

Although Grounding DINO can carry out open-vocabulary object detection task, it still faces some issues. There are primarily two problems. First, the core step of Grounding DINO requires a noun as input to locate the position of that noun in the image. Moreover, it introduces methods to extract a series of nouns from a sentence description, enabling it to perform object detection tasks. However,

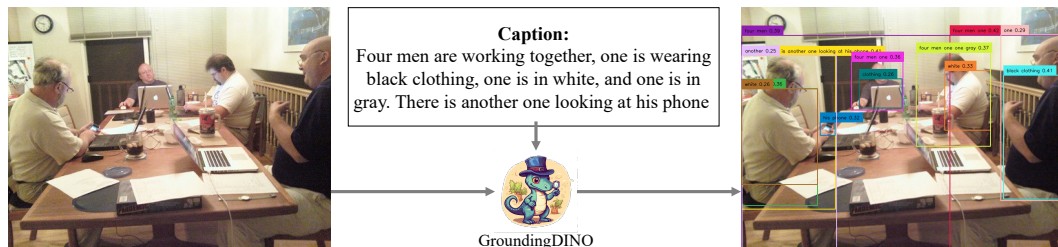

Figure A6: **An example of Grounding DINO undertaking an open-vocabulary task**, where it encounters issues with ambiguous labels and faces challenges in distinguishing between different individuals within the same category.

the method of extracting nouns can sometimes err, leading to Grounding DINO producing some bizarre labels. For example, as illustrated in Figure A6, Grounding DINO outputs ambiguous labels such as "one", "four men one one gray", "another".

The second issue, which is more severe, is Grounding DINO's difficulty in distinguishing between different individuals of the same category. As shown in Figure A6, although Grounding DINO identifies four people, it is challenging to determine which individual is represented by which bounding box with vague labels like "four men one". Note that the ASSIST benchmark serves as such a complex benchmark, incorporating numerous scenarios that more closely mirror real-life situations where it is necessary to distinguish different objects within the same or similar categories.

Benefiting from ASSIST's powerful capabilities, Grounding DINO can overcome these two issues with the aid of ASSIST. For the first problem, ASSIST inherently possesses the ability to identify important objects in an image, allowing Grounding DINO to receive a list of objects from ASSIST, resulting in a more accurate and comprehensive list of nouns. Regarding the second issue, as introduced in **??**, by utilizing the list of objects provided by ASSIST, along with detailed descriptions of each object, it is possible to post-process Grounding DINO's predictions. This enables the precise distinction of different individuals within the same category label.

### A.5.2 POINT QUESTION ANSWERING

**Method of applying ASSIST.** In our experiment, PointQA is designed to answer questions related to image regions based on the description of the image. Most descriptions provided by Visual Language Models (VLMs) cannot accomplish this task as their descriptions lack positional information. However, ASSIST provides both the positional information of objects within the image and their corresponding descriptions. Given a target area, by combining descriptions of different objects based on their positional relationships, one can create a description relevant to the location. Specifically, as illustrated in Figure A3, we compute the Intersection Over Union (IOU) between the target area and the positions of all objects. By combining the descriptions of objects with high overlap, we obtain a description that is closely related to the target area. Then, we feed this description to the question-answering model to answer the question.

### A.5.3 POINTING QUESTION ANSWERING

**Method of applying ASSIST.** The PointingQA task requires selecting the most appropriate region from a set of candidate areas based on a textual prompt. VLMs struggle to complete this task because they often lack the ability to perceive input location information. However, since ASSIST decomposes image descriptions into a series of basic elements, each with its corresponding location, we can leverage this feature to accomplish the task. As shown in Figure A3, the method is divided into three steps. First, we calculate the CLIP similarity between each object's description and the input textual prompt, obtaining scores for each object. The more relevant an object is to the text description, the higher its score. Secondly, we calculate scores for each candidate region by weighting the sum of object scores based on the overlap between the candidate region and the object's location. The greater the overlap with the candidate area, the larger the proportion of that object's score. In the third step, the region with the highest score is selected as the answer.

### A.5.4   IMAGE GENERATION

**Method of enhancing SDXL by ASSIST.** Even as one of the most renowned models for text-to-image generation, SDXL often struggles to understand complex prompts and generate precise images accurately. This is primarily because SDXL employs CLIP for text understanding, which limits its ability to comprehend the text. However, each basic element within a complex prompt is not complicated for SDXL to understand and generate. Therefore, by breaking down complex texts into basic elements, ASSIST can significantly assist SDXL in simplifying complex tasks. Specifically, SDXL can first create the background, then sequentially generate each object, and finally assemble the different parts. Currently, there are many methods that can be utilized for image stitching, such as Anydoor (Chen et al., 2024), Collage Diffusion (Sarukkai et al., 2024), *etc.* Sometimes, images can also be directly stitched together and then refined using SDXL as the base model, with SDEdit (Meng et al., 2021) for refining the images, but this typically requires the images to be relatively simple. Aside from generating individual parts of the image and then stitching them together, another approach is to sequentially inpaint (Rombach et al., 2022) objects onto the image using inpainting methods.

**More results.** We provide more examples in Figure A13

### A.5.5   PRECISION & RECALL AND USER STUDY

When calculating precision and recall, it involves identifying which objects have been predicted by different captioners. For other captioners, this can be challenging because directly extracting nouns would include many nouns that cannot be considered objects. Therefore, we utilize VLMs to accomplish this task. Specifically, we input the model's captions into the VLMs, requesting them to extract the important objects contained within. For LLAVA(ASSIST)-CAPTIONER, this process is straightforward because ASSIST explicitly provides a list of objects. This also highlights the advantages of ASSIST.

### A.5.6   ASSIST ON VIDEO CAPTIONING

We provide examples (as Figures A14 to A16) as a supplementary of the main paper.

**Overall description**:

    **Style**: 'The image is a photograph with a realistic style.'

    **Theme**: 'The theme of the image is transportation, specifically a train traveling through a rural landscape.'

    **Background description**: 'The background of the image features a rural landscape with elements of nature and infrastructure. There is a bridge with green metal railings crossing over the train tracks. Beyond the bridge, a fence made of wooden posts and rails encloses a field. The field appears to be grassy with some patches of bare earth. The sky is overcast, with a pale, diffused light suggesting an overcast or cloudy day.'

    **Foreground description**: 'In the foreground, a train is captured in motion on the tracks. The train is painted in a blue and yellow color scheme. The train has multiple carriages, and the windows reflect the surrounding environment. The tracks are made of steel rails with wooden sleepers, and they run parallel to a grassy embankment on the left side of the image.'

**Object list**:

    Train:

        Category: inanimate, foreground

        Description: 'The <train> 's body is long and sleek, with <windows> lined along its side. The front <car> has a curved nose with a destination <sign> and <headlights>. The <train> is composed of several <carriages> connected together.'

        Color: blue and yellow

        Position: [200, 160, 441, 367]

    Track:

        Category: inanimate, foreground

        Description: 'The <track> consists of parallel <steel rails> supported by wooden <sleepers>. It stretches into the distance, guiding the <train>.'

        Color: rusty brown rails, brown sleepers

        Position: [128, 112, 553, 425]

    Bridge:

        Category: inanimate, background

        Description: 'The <bridge> spans over the <tracks> with a structure made of metal <beams> and <railings>. It appears functional and unadorned.'

        Color: green railings

        Position: [54, 95, 271, 160]

    Fence:

        Category: inanimate, background

        Description: 'The <fence> is constructed of wooden <posts> and <rails>, enclosing the <field> and providing a boundary.'

        Color: natural wood tone

        Position: [274, 137, 638, 184]

    Field:

        Category: inanimate, background

        Description: 'The <field> is predominantly grass-covered, with some areas of bare <soil>. It is bordered by the <fence> and <trees>.'

        Color: green grass, brown soil

        Position: [283, 161, 638, 421]

    Tree:

        Category: inanimate, background

        Description: 'The <trees> have bare branches, indicating a lack of <leaves> which could suggest a seasonal change.'

        Color: dark brown branches

        Position: [207, 0, 404, 146]

    Sky:

        Category: inanimate, background

        Description: 'The <sky> is overcast, with a uniform light grey color, suggesting cloudy weather.'

        Color: light grey

        Position: [1, 0, 636, 103]

**Relationship**:

    <Train> [is traveling on] <track>

    <Train> [is passing under] <bridge>

    <Bridge> [spans over] <track>

    <Fence> [encloses] <field>

    <Field> [is bordered by] <tree>

    <Field> [is bordered by] <Fence>

    <Tree> [is standing in] <Field>

Figure A7: **A complete example of ASSIST.**

**Captioner&Grounding**

**Overall description**:

**Style**: 'The image is a candid, action-oriented photograph.'

**Theme**: 'The theme of the image is water sports, specifically surfing.'

**Background description**: The background of the image is dominated by the ocean with its varying shades of blue and white due to the churning waves and foam. The water appears dynamic and powerful, suggesting a strong surf. There is no clear horizon line, but the waves occupy the majority of the background space, creating a sense of immersion in the ocean environment. The light seems natural, likely from the sun, and it enhances the texture of the water and waves. The ocean is the only element in the background, emphasizing the vastness and the solitary nature of the surfing activity.'

**Foreground description**: 'In the foreground, a man is captured in the midst of surfing a wave. He is wearing a cap, shorts, and is shirtless, which indicates a warm climate or season. The man is in a semi-crouched position on a surfboard, navigating the wave with a paddle in his left hand. His facial expression shows concentration and determination. The surfboard is mostly white with a design on its surface, and it cuts through the water, leaving a trail of spray behind. The man's musculature and stance suggest that he is experienced and in control of the surfboard. The water around the surfboard is turbulent, with foam and spray being generated by the movement of the surfboard and the wave.'

**Object list**:

Man:
> Category: living, foreground
> Description: 'The <man>'s <torso> is bare, and he is wearing a <cap> on his <head>. His lower body is covered by <shorts>. He is in a semi-crouched position on the <surfboard>, holding a <paddle> in his left hand. His facial expression shows focus.'
> Color: skin tone, green shorts, white cap
> Position: [200, 201, 270, 288]

Surfboard:
> Category: inanimate, foreground
> Description: 'The <surfboard> is under the <man>, supporting him as he rides the <wave>. It has a design on its surface and is cutting through the <water>.'
> Color: predominantly white with a design
> Position: [178, 285, 283, 326]

Paddle:
> Category: inanimate, foreground
> Description: 'The <paddle> is held by the <man> in his left hand, assisting him in navigating the <wave>.'
> Color: black shaft, white blade
> Position: [103, 171, 266, 271]

Wave:
> Category: inanimate, background
> Description: 'The <wave> is large and powerful, with <water> churning and creating <foam> and <spray> as it breaks.'
> Color: shades of blue and white
> Position: [2, 63, 638, 422]

Ocean:
> Category: inanimate, background
> Description: 'The <ocean> fills the background, characterized by its dynamic <waves> and <foam>.'
> Color: various shades of blue
> Position: [2, 2, 638, 424]

**Relationship**:
> <Man> [is riding] <Surfboard>
> <Man> [is holding] <Paddle>
> <Surfboard> [is cutting through] <Wave>
> <Wave> [is breaking around] <Man>
> <Man> [is surfing on] <Ocean>
> <Ocean> [is supporting] <Surfboard>

Figure A8: **A complete example of ASSIST.**

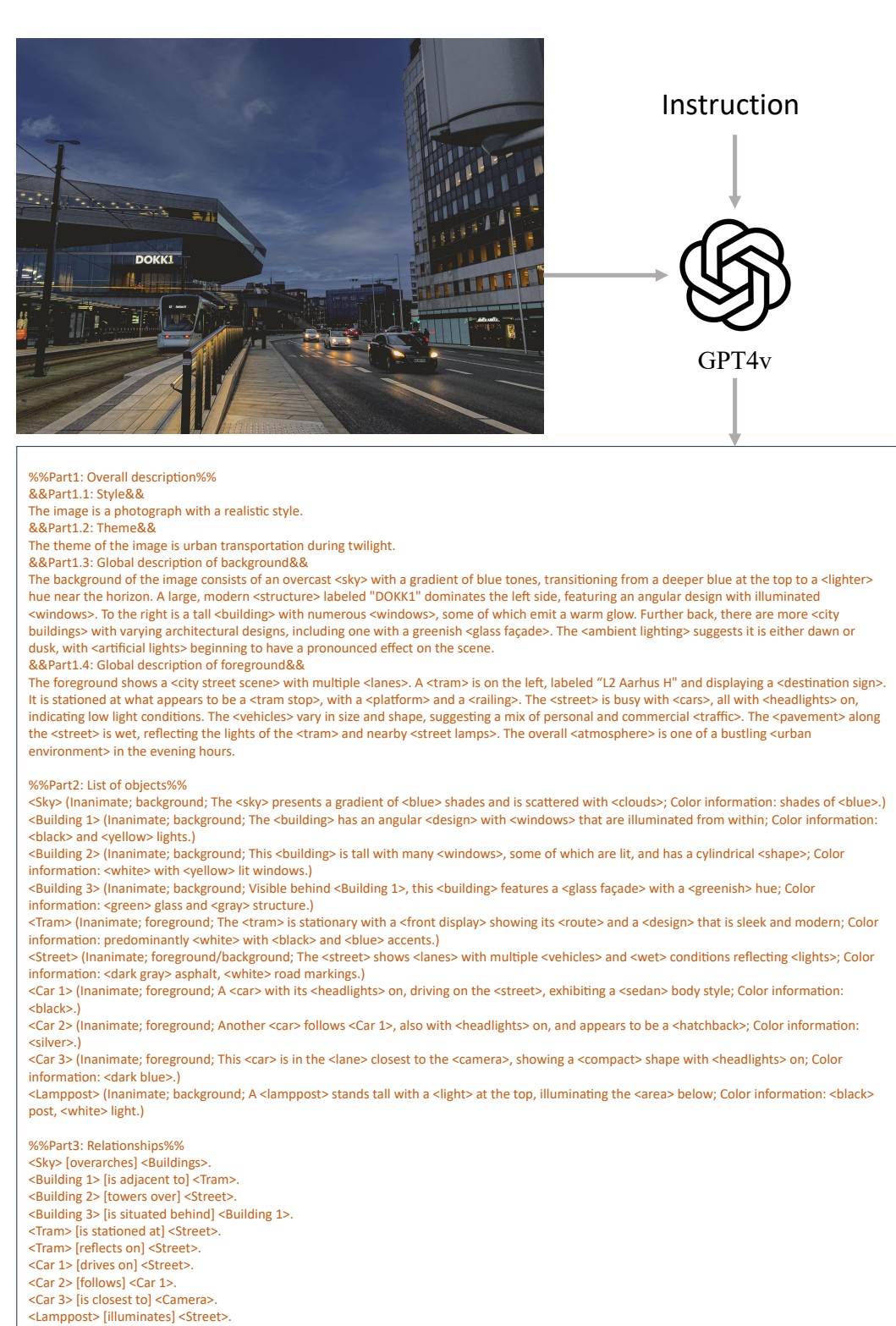

Figure A9: An example of ASSIST in string format obtained by GPT-4V.

1   Hello, I would like to ask for your help in describing an image. Please note that I would like the description to be as
2   detailed as possible. Please strictly respond following my instructions and do not print any redundant words.

3   This description needs to include three parts. The title of each part should be '%%Part1: Overall description%%', '%%Part2:
4   List of objects%%', and '%%Part3: Relationships%%'. All important nouns in your response have to be bounded by '<' and
5   '>'!

6   The first part is an overall description of the image. Your answer to this part should consist of three parts, one sentence to
7   describe the style of the image, one sentence to describe the theme of the image, and several sentences to describe the
8   image. The titles of these parts are '&&Part1.1: Style&&', '&&Part1.2: Theme&&', '&&Part1.3: Global description of
9   background&&', 'Part1.4: Global description of foreground&&'. The global description should be as detailed as possible
10  and at least 150 words in total. If there is text content in the image, you can also describe the text, which should be bound
11  by quotation marks. All important nouns in your response have to be bounded by '<' and '>'!

12  The second part is to list all the objects in the image, as many as possible, in order of importance. Note that any object
13  should not be a part of other objects. Note that the listed object should not be the plural. If there are multiple individuals
14  of the same category of objects, please list them separately. For example, if there are three apples in the picture, they
15  should be listed as 'Apple 1,' 'Apple 2,' and 'Apple 3.', respectively. Additionally, the objects should be classified into two
16  categories: living and inanimate objects. Living refers to creatures such as humans, cats, dogs, and plants, while other
17  lifeless objects belong to the category of inanimate objects. Finally, each object should have a very detailed description,
18  with more important objects receiving more detailed descriptions. Each description should be at least 30 words and the
19  important nouns in it have to be bounded by '<' and '>'. You should also identify whether this object belongs to the
20  foreground or background. You should additionally provide a sentence to describe the color information of the object.
21  Therefore, the format for listing each object should be 'Object Name (Category (Living/Inanimate);
22  foreground/background; Description; Color information)'. Specifically, the detailed description of an object should focus
23  on its part and its action. All descriptions should be in the forms of, object's + part + verb + object/adjective or object + is +
24  present participle. The description should be detailed as well as possible, and try to describe all parts of this object. You
25  should specifically notice if there is a sky, tree, sun, or other object in the background of the environment. All important
26  nouns in your response have to be bounded by '<' and '>'!

27  The third part is to describe the relationships between all the objects in pairs. Please list them one by one. Additionally,
28  please describe the relationship between object A and object B in the format of 'Object A' + 'Action' + 'Object B.' Please
29  don't print the same relation twice. For example, if there is "A relation B", you shouldn't print 'B relation A' again. All
30  important nouns in your response have to be bounded by '<' and '>'!

31  I will provide you with an example of the last two parts of a description to show you the desired format. You should only
32  focus on the format of this example instead of the content of it. You should use the same format to respond.

33  "%%Part2: List of objects%%
34  <Woman> (Living; foreground; The <woman>'s <hair> is bundled in a <scarf>. Her <torso> is covered with a <black shirt>.
35  Her <lower body> is clad in <blue jeans>. Her <legs> move through the <water>. Her <right hand> holds a pair of <shoes>;
36  Color information: <black> shirt, <blue> jeans, <orange> scarf.)
37  <Water> (Inanimate; foreground/background; The <water> floods the <street>, reflecting the <sky> and <surrounding
38  objects>; Color information: <murky blue-grey>.)
39  <Building 1> (Inanimate; background; The <building> has a <façade> with <doors> and <windows>, showing signs of
40  <water damage>; Color information: <pale yellow>.)
41  <Building 2> (Inanimate; background; This <building> is similar to <Building 1> but with a <red> roof visible above the
42  <flood>; Color information: <light orange> walls, <red> roof.)
43  <Vehicle 1> (Inanimate; background; A <vehicle> is partially submerged, showing only the <roof> and <upper parts>; Color
44  information: <white>.)
45  <Vehicle 2> (Inanimate; background; Another <vehicle>, also partially submerged, with a <visible logo>; Color information:
46  <silver>.)
47  <Sky> (Inanimate; background; The <sky> is filled with <clouds>, implying recent or ongoing <precipitation>; Color
48  information: <gray>.)
49  %%Part3: Relationships%%
50  <Woman> [is walking through] <Water>.
51  <Woman> [is moving away from] <Camera>.
52  <Water> [reflects] <Sky>.
53  <Water> [surrounds] <Vehicles>.
54  <Buildings> [line] <Street>.
55  <Vehicle 1> [is submerged by] <Water>.
56  <Vehicle 2> [is submerged by] <Water>.

Figure A10: **The instruction** for GPT-4V to obtain ASSIST from an image. We highlight the parts involving specific examples in orange.

A cat and a dog gather around a pizza on the highway. → ASSIST-Captioner

**Overall description**

**Style**: The image is a candid, real-life photograph with a humorous setup.

**Theme**: The theme of the image is an unexpected interaction between animals and human food.

**Background description**: The background of the image features a highway setting with a large, dark-colored **rock** on the left side and a **scrapped car** on the right. The sky is overcast, with a dull, uniform gray tone suggesting an overcast or cloudy day. The **highway** itself is made of asphalt and has visible lane markings. The **rock** and **car** appear to be out of place on the **road**, indicating a possible scene of an accident or roadside debris. There are no visible people or moving vehicles, which adds to the stillness of the scene.

**Foreground**: In the foreground, a **cat** and a **dog** are positioned on the road, closely inspecting a **pizza** that lies between them. The **cat**, on the right, is leaning forward with its head down, sniffing or showing interest in the **pizza**. The **dog**, on the left, mirrors the **cat**'s posture, with its nose also directed towards the **pizza**. The **pizza** itself is large, with visible toppings that include cheese and pepperoni. The animals' proximity to the **pizza** and their focused attention on it create a humorous and unexpected scene, as it is unusual to see wild animals interacting in such a human-like manner. The **road** surface is a faded gray, and the animals' fur appears well-defined against this backdrop.

**Object list**

① *Name:* **Cat**; *Category:* Animal, foreground; *Description:* The cat's body is arched slightly as it leans over the pizza. Its head is lowered towards the pizza, its eyes seem to be focused on the food, and the cat's fur is dark blue; *Color*: dark blue; *Position:* [677, 178, 981, 361]

② *Name:* **Dog**; *Category:* Animal, foreground; *Description:* The dog is standing with its body angled towards the pizza. Its head is also lowered to get a closer look at the pizza. The dog has fur that appears to be short and smooth; *Color:* dark gray; *Position:* [45, 167, 375, 423]

④ *Name:* **Pizza**; *Category:* Inanimate, foreground; *Description:* The pizza is round with a crust and several toppings that include cheese and pepperoni. It is placed directly on the road surface; *Color:* golden crust, red pepperoni, yellow cheese; *Position:* [428, 183, 632, 324]

⑤ *Name:* **Rock**; *Category:* Inanimate, background; *Description:* The rock is large and irregularly shaped, with a rough texture and a dark color that contrasts with the asphalt of the highway; *Color:* dark gray; *Position:* [281, 537, 488, 649]

⑥ *Name:* **Scrapped Car**; *Category:* Inanimate, background; *Description:* The scrapped car is heavily damaged, with twisted metal. It is positioned on its side, suggesting it was involved in a severe accident; *Color:* rusty and metallic tones; *Position:* [584, 478, 852, 655]

**Relationship**

① < Dog > on the left side of < pizza >   ② < Cat > on the right side of < pizza >
③ < Pizza > is lying on < highway >   ④ < Rock > is blocking < highway >
⑤ < Rock > is adjacent to < Scrapped Car >   ⑥ < Scrapped Car > is blocking < highway>

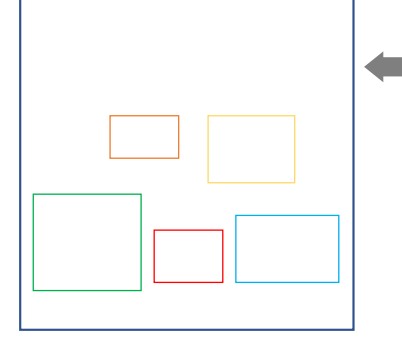

Figure A11: **An example of how LLAVA(ASSIST)-CAPTIONER transforms a brief prompt into the distinctive style of ASSIST.**

Under the bright moon, there is an icy sky and snow. In the center of the scene, a bonfire blazes. To the left of the bonfire are an arctic fox and a polar bear, to the right is a penguin, and above the bonfire flutters a small ice butterfly.

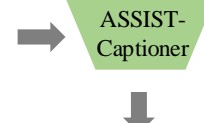

ASSIST-Captioner

### Overall description

**Style**: The image is a color photograph with a clear focus and a balanced composition.

**Theme**: The theme of the image is a serene, wintery scene featuring various arctic animals gathered around a warm bonfire.

**Background description**: The background of the image is mainly the bright moon in the upper part of the screen. Below the sky, the horizon is lined with a layer of snow that blankets the ground, reflecting the light from the bonfire and creating a contrast with the dark sky. The snow appears to be undisturbed except for the area immediately around the fire, where it is melted.

**Foreground**: In the foreground, the central focus is the **bonfire**, with vibrant flames leaping upwards, casting a warm glow on the surrounding snow. The fire's light illuminates the figures of the animals and the snow around them. To the left of the fire, an **arctic fox** sits attentively, its fur appearing thick and white. To the right, a **polar bear** sits with a relaxed posture, its white fur contrasting with the darker elements of the scene. In front of the fire, a **penguin** stands out with its black and white coloring, with its back facing the camera. Above the fire, an **ice butterfly** with outstretched wings adds a whimsical touch to the scene. The animals are positioned on a flat surface, likely a frozen body of water, as indicated by the presence of the ice butterfly and the reflection of the fire on the snow.

### Object list

① *Name:* **Arctic Fox**; *Category:* Animal, foreground; *Description:* The arctic fox's body is facing the camera, with its head turned slightly to the side. Its fur is thick and white; *Color:* White; *Position:* [153, 235, 337, 419]

② *Name:* **Polar Bear**; *Category:* Animal, foreground; *Description:* The polar bear is sitting with its body oriented away from the camera. Its fur is thick and white, and its head is turned to the side, giving a profile view; *Color:* White; *Position:* [266, 264, 481, 627]

③ *Name:* **Penguin**; *Category:* Animal, foreground; *Description:* The penguin stands with its body turned away from the camera, its head slightly tilted. It has black back and white belly; *Color:* Black back, white belly; *Position:* [471, 154, 614, 416]

④ *Name:* **Bonfire**; *Category:* Inanimate, foreground; *Description:* The bonfire consists of flames leaping upwards from a pile of wood. The fire is bright and orange, casting light on the surrounding snow; *Color:* Orange flames, brown and black wood; *Position:* [307, 204, 614, 409]

⑤ *Name:* **Ice Butterfly**; *Category:* Animal, foreground; *Description:* The ice butterfly is positioned above the bonfire, with its wings spread wide. It appears to be made of ice and is translucent with a white hue; *Color:* Translucent with a white tint; *Position:* [378, 768, 450, 896]

### Relationship

① < Arctic Fox > on the left side of < Bonfire >      ② < Arctic Fox > sits near < Polar Bear>

③ < Polar Bear > on the right side of < Arctic Fox >   ④ < Penguin > in front of < Bonfire >

⑤ < Ice Butterfly > floats above < Bonfire >      ⑥ < Polar Bear > sits beside < Bonfire>

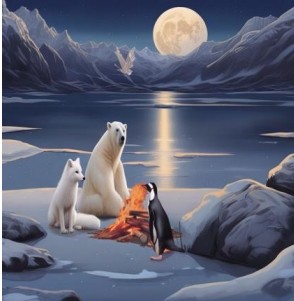

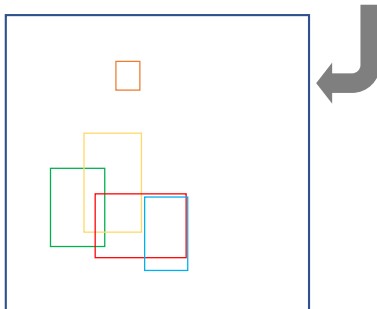

Figure A12: **An example of how LLAVA(ASSIST)-CAPTIONER transforms a relatively complex prompt into the distinctive style of ASSIST.**

| Caption | ASSIST + SDXL | DALL-E 3 | SDXL |
|---|---|---|---|

In the deep sea, an abandoned large ship full of marine life sank to the bottom of the sea. There are two blue balloons floating in front of the ship. There is a dolphin swimming below the balloon. There is a drifting bottle floating in the deep sea, inside which is a sailboat

In a yoga studio, there is an artwork of a green jade dragon, with a white cat lying on the right side of the artwork. On the distant ground, against the wall, there is a painting depicting war

In an abandoned factory building, sunlight filtered in. A technologically advanced spaceship flies over the factory building. Listening to a motorcycle below the spaceship, there is a pink guitar on the ground to the right of the motorcycle.

On a pink night, there was a pool in the center of the lawn, and a purple sports car was floating on the pool. There was a light bulb on the hood of the sports car, and there was an orange goldfish in the bulb. On the left side of the car is a small, colorful robot

There is a small river in the forest, and there is a stone bridge on the river. There is a golden praying mantis on the bridge. There is a mongoose standing by the riverbank, and to its right lies a turtle

In an old-fashioned subway station, there is a emerald green lion, a gray white wolf, and a colorful paper crane standing together waiting for the subway

Figure A13: **Additional examples of ASSIST on image generation.**

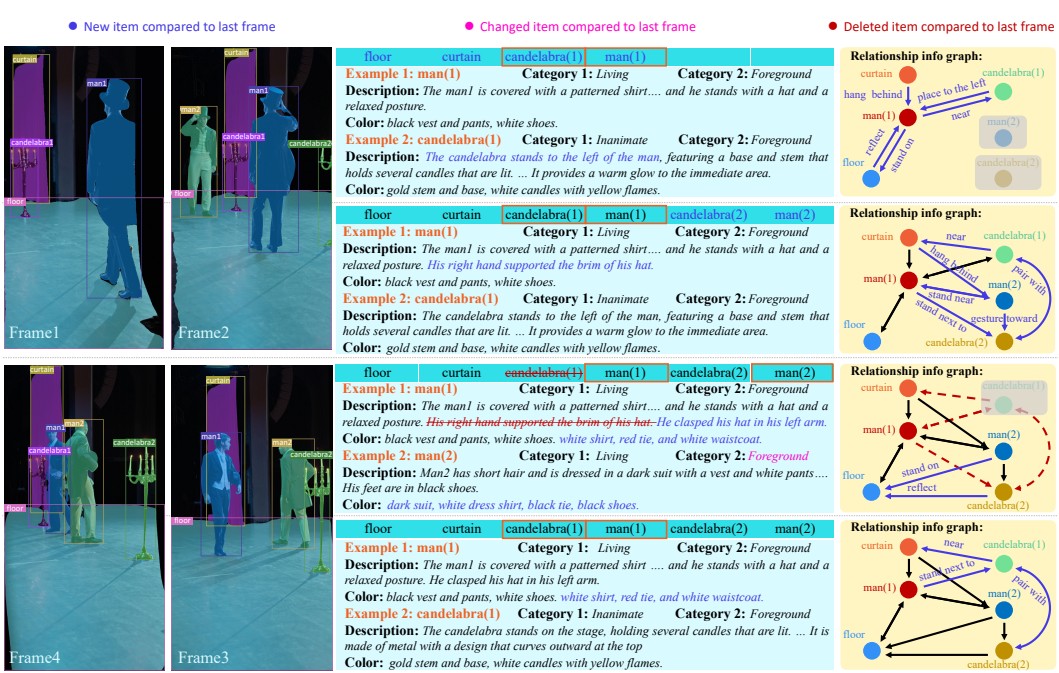

Figure A14: **An example of ASSIST on video captioning**, which includes three components: an overall description, an object list, and their relationships, each dynamically evolving over time. With respect to a prior frame, updates are color-coded: new elements in green, removed in red, altered in gold, and persistent ones in black. ASSIST thus adeptly captures the temporal changes and salient details of each video frame, while its structured nature potentially aids in downstream model comprehension.

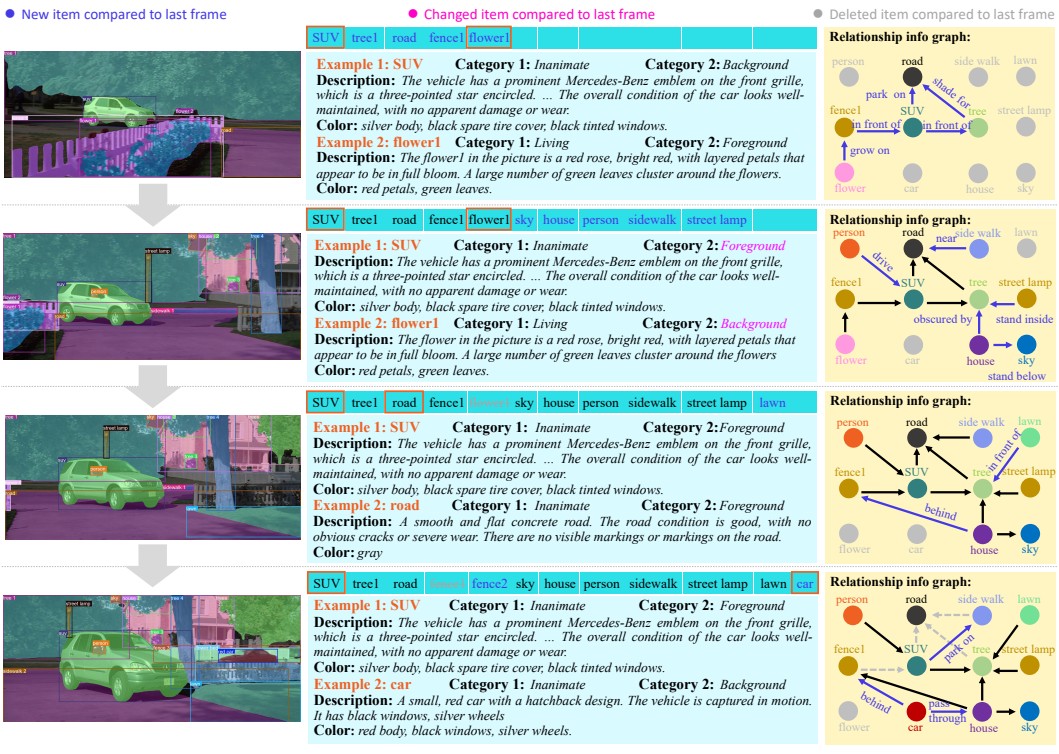

Figure A15: **An additional example of ASSIST on video captioning**.

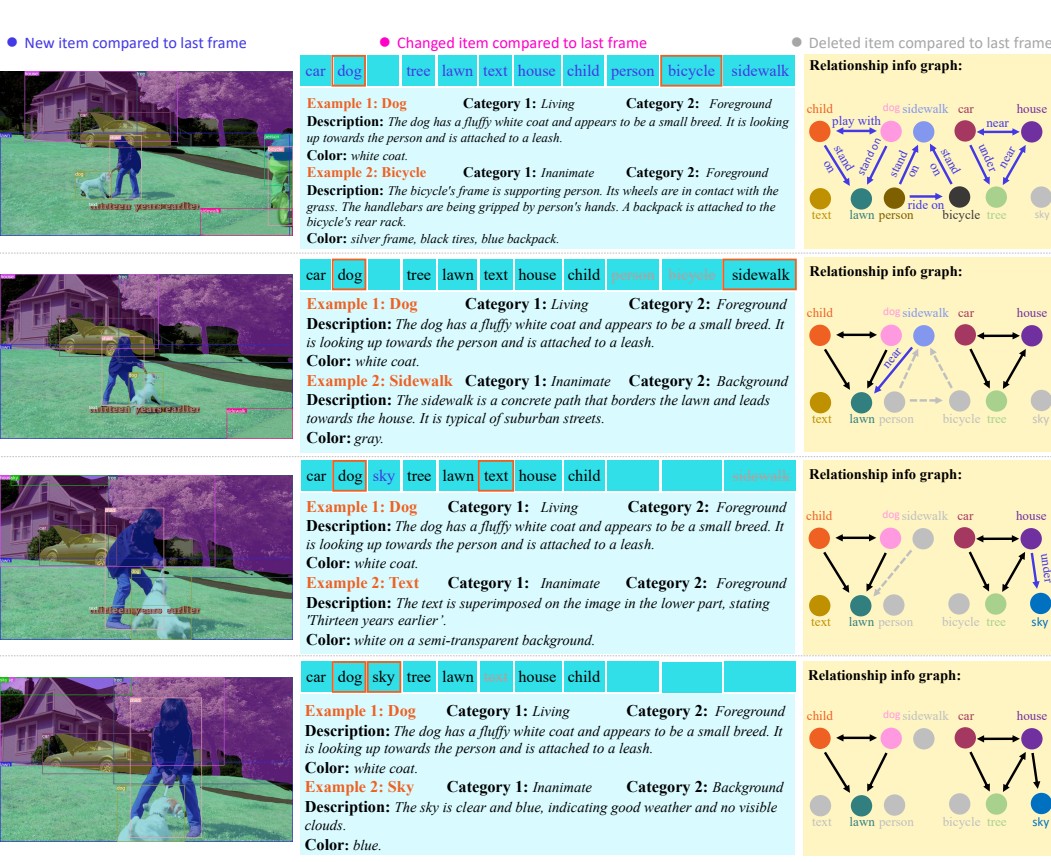

Figure A16: **An additional example of ASSIST on video captioning**.

