# OpenReview forum: "Asking Specifically Instead of Ambiguously to Your GPT Improves Image Caption"
_ICLR.cc/2025/Conference — ICLR 2025 Conference Withdrawn Submission_

### Official Review · Reviewer_rRHp · 2024-11-03

**Soundness:** 3
**Presentation:** 3
**Contribution:** 1
**Rating:** 6
**Confidence:** 2

**Summary:**

This paper studies how to improve image description quality from GPT using prompt with detailed questions and specific structure, which significantly enhances the attention of VLMs to important objects. The high quality annotations are used to fine tune LLAVA, for object detection and description, which is later used for object grounding with the assist from other vision foundation model.

**Strengths:**

- simple to implement, just call GPT with proper instruction
- supports multiple task, such as image captioning, open-vocabulary object detection and image generation
- significant improvement on object detection task as shown in Table 2

**Weaknesses:**

- lack of technical innovation, based on pure prompt engineering and combination with other vision foundation models, ie, grounding dino, internVL, clip.
- the improvement is based on learning from GPT-V, which is an unfair comparison with other VLLMs
- No evaluation on object relationship accuracy, only indirectly shown through the help on T2I attributes accuracy, Table 4.

**Questions:**

- does this approach work if instead of using GPT-4V, use opensource VLLMs to get initial dense caption?
- Although the method performs well on object and relationship detection, but it does not help VQA much as shown in Table A1. Is it because the full description of the objects can distract the VQA model from focusing on the most important objects?

---

### Official Review · Reviewer_oWB1 · 2024-11-03

**Soundness:** 2
**Presentation:** 3
**Contribution:** 2
**Rating:** 5
**Confidence:** 2

**Summary:**

The paper introduces a method designed to improve image captioning in VLMs by using specific, element-focused questions instead of vague prompts. The method decomposes the task into structured, targeted questions that enhance the model’s focus on essential elements, thereby improving object recognition and attribute description. This method was validated with a new dataset  and a fine-tuned LLAVA model, achieving significant improvements in object recognition, caption precision, and recall on benchmarks. The method also demonstrated advantageous for downstream applications, such as open-vocabulary object detection and image generation.

**Strengths:**

1.The proposed method, breaking down image captioning into specific, targeted questions rather than relying on broad, ambiguous prompts, is reasonable.

2. The method significantly improves performance metrics, with notable gains in object recognition, caption precision, and recall.

3. The authors support their claims with comprehensive analyses across various benchmarks. Detailed performance comparisons substantiate the method's enhancements in accuracy and robustness.

**Weaknesses:**

The technical contribution of this paper appears limited. The core novelty—using specific, element-focused questions to improve caption generation—is not particularly surprising, as similar techniques are already common among practitioners. For instance, the concept of using chain-of-thought prompts to create detailed captions has been discussed in practical settings, as seen in this blog: https://docs.llamaindex.ai/en/stable/examples/multi_modal/gpt4v_experiments_cot/. Another relevant example is the Set-of-Mark Prompting method: Set-of-Mark Prompting Unleashes Extraordinary Visual Grounding in GPT-4V, which shares some similarity with the proposed method.

**Questions:**

Has the author observed any bias, hallucination in the GPT generated answer? If yes, is there estimate or evaluation of the accuracy of the GPT generated answers?

---

### Official Review · Reviewer_b2uq · 2024-11-03

**Soundness:** 3
**Presentation:** 3
**Contribution:** 3
**Rating:** 6
**Confidence:** 4

**Summary:**

However, current VLM-based image captioning methods often miss important details, recognize incorrect objects or relationships, and deliver suboptimal captions for downstream applications.
One primary reason for this issue is the ambiguous prompts typically used, such as ”describe this image in detail,” which fail to guide the VLM’s focus on specific elements within the image.
To address this, the authors extensively explore the difference between using ambiguous prompts and decomposing them into a series of specific questions.
The authors find that asking a series of targeted element-specific questions significantly enhances the attention of VLMs to important objects, the consistency of the answers under repeated questions, and the alignment with their training data distribution.
Building on this insight, the authors introduce ASSIST, a method that systematically decomposes image caption prompts into a sequence of focused questions corresponding to distinct image elements.
They annotated 100k images using GPT- 4V with this approach and fine-tuned a LLAVA model, resulting in a captioner that greatly improves caption accuracy and quality.

**Strengths:**

- The idea to generate specific prompts for caption generation is novel.
- The proposed method shows noticeable performance improvements compared to the previous works.

**Weaknesses:**

- It would be better to have a more detailed discussion on the prompts for LLAVA. In Figure 5, in the prompt for LLAVA, it says, “Does it correct?” which is grammatically incorrect ("Is it correct?"). How did the authors decide the prompts and how does the result change if the prompts differ?
- Also, the experiments on detailed design choices for the proposed method are missing. For example, when extracting the object list, how does the final performance change when the object detector changes? It would also be essential to analyze the performance of ASSIST by changing the list of descriptions to use or the number of objects, etc.

**Questions:**

Please refer to the questions in the weakness.

---

### Official Review · Reviewer_V5yw · 2024-11-03

**Soundness:** 2
**Presentation:** 2
**Contribution:** 2
**Rating:** 5
**Confidence:** 4

**Summary:**

The proposed method, ASSIST, is a form of prompt engineering where the instruction/prompt is more specific than the commonly used prompt of "Please describe the image in detail". The authors also propose a dataset using the ASSIST framework called ECO, when VLMs like LLaVA and open vocabulary models are fine-tuned on this dataset, they outperform prior models on wide range of downstream tasks.

**Strengths:**

1. The paper is well written, the setup and motivations are clearly described.
2. The introduced method and the dataset have a positive effect when used to fine-tune VLMs and other open vocabulary methods.

**Weaknesses:**

1. Prior works like [1] have shown that existing VLMs do not perform well in understanding attributes and relationship between various entities in the scene. It is not clear from the manuscript on how this challenge is overcome, especially in building the coarse scene graph, examples of which is shown in the appendix. Is it possible to evaluate VLMs using prompt engineering in ASSIST framework on the MMVP benchmark? Also, it would be good to understand how VLMs trained on ECO dataset perform on this benchmark.
2. Results on ARO and SugarCrepe benchmarks would be nice to have to evaluate if the introduced datasets can enable the model to reason about compositionallity of various entities in the scene.
3. It would also be good to compare the dataset produced by ASSIST against VLM finetuned/trained using other densely captioned datasets like ReCAP-DataComp[2] or ReCAP datasets introduced in LLaVA-NeXT[3].

[1] - Eyes Wide Shut? Exploring the Visual Shortcomings of Multimodal LLMs
[2] - What If We Recaption Billions of Web Images with LLaMA-3?
[3] - LLaVA-NeXT: What Else Influences Visual Instruction Tuning Beyond Data?

**Questions:**

It would be good to understand the benefits of the proposed dataset on benchmarks that identify shortcomings of VLMs like ARO, SugarCrepe & MMVP as mentioned in the weaknesses above. I am willing to reconsider my score based on the outcome on these datasets.

---

### Official Review · Reviewer_HnQt · 2024-11-04

**Soundness:** 2
**Presentation:** 2
**Contribution:** 2
**Rating:** 5
**Confidence:** 4

**Summary:**

The paper presents an approach to better prompt VLMs to obtain descriptive captions: instead of ambiguous prompts like ”describe this image in detail,” the study proposes to use a series of specific, element-focused questions, with the help of various external tools. The study then annotates 100k images in this approach using GPT-4V, and finetunes a LLava model for captioning. Various evaluation methods (caption text quality, for detection, and for generation) are adopted to validate the improvement in caption quality.

**Strengths:**

1. The study comprehensively designs and adopts various evaluation methods to validate the quality of the generated descriptive captions, including caption-QA, for object detection, as T2I conditions, and so on. The evaluation would be beneficial for various scenarios that require descriptive long caption evaluation.

2. The study explores an effective way to combine the main VLM with other tools to obtain better quality descriptive long captions. This tool would be beneficial for various applications that need prompting VLMs for image description.

**Weaknesses:**

1. The experiment setting is limited to distilling GPT-4V to a weaker model (LLava). To better support the claim that ASSIST is a better way for obtaining captions, there should be a llava-generated version of the Enumerate Common Objects in Context (ECO) dataset, showing it performs better than the naive LLava outputs, as well as further finetuning improves beyond the LLava baseline.

2. A few important model variants are missing in experiment comparison as baselines. For LLava finetuning, instead of comparing with LLava or other VLMs without descriptive caption finetuning, there should be a comparison between LLava-ECO dataset finetuned, and LLava finetuned with also 100k GPT-4V generated captions prompted with ambiguous prompts like ”describe this image in detail” (e.g., using shareGPT4V). This could support LLava-ASSIST is improved because of the better caption quality in ECO dataset, instead of seeing more long captions.

3. For the motivation that the model fails on instructions like ”describe this image in detail”, this might be a problem in existing models, specifically in the instruction tuning stage, instead of a universal property.

4. The prompting framework looks interesting, especially the grounding part. Extra discussions and comparisons are needed to differentiate it from previous works that use external tools to generate grounded description data, e.g., “Visual Clues: Bridging Vision and Language Foundations for Image Paragraph Captioning.”

**Questions:**

1. Line 71: “Your GPT”
2. Line 1052, reference broken

---

### Note · Authors · 2024-11-15

I have read and agree with the venue's withdrawal policy on behalf of myself and my co-authors.